# Modular interneuron circuits control motion sensitivity in the mouse retina

Andrew Jo[1,2], Sercan Deniz[1,2], Suraj Cherian[1], Jian Xu[1], Daiki Futagi [1], Steven H. DeVries [1] & Yongling Zhu [1] ✉

Neural computations arise from highly precise connections between specific types of neurons. Retinal ganglion cells (RGCs) with similar stratification patterns are positioned to receive similar inputs but often display different response properties. In this study, we used intersectional mouse genetics to achieve single-cell type labeling and identified an object motion sensitive (OMS) AC type, COMS-AC(counter-OMS AC). Optogenetic stimulation revealed that COMS-AC makes glycinergic synapses with the OMS-insensitive HD2p-RGC, while chemogenetic inactivation showed that COMS-AC provides inhibitory control to HD2p-RGC during local motion. This local inhibition, combined with the inhibitory drive from TH2-AC during global motion, explains the OMS-insensitive feature of HD2p-RGC. In contrast, COMS-AC fails to make synapses with W3(UHD)-RGC, allowing it to exhibit OMS under the control of VGlut3-AC and TH2-AC. These findings reveal modular interneuron circuits that endow structurally similar RGC types with different responses and present a mechanism for redundancy-reduction in the retina to expand coding capacity.

It is widely acknowledged that diverse local interneuron populations can confer diverse signaling properties on nearby projection neurons. Yet, at a higher level, regions of the brain that contain diverse interneurons are thought to be functionally specialized, such as the color and motion processing areas of the visual cortex. In the retina, the responses of more than 40 types of retinal ganglion cells (RGCs)[1–5], the retinal projection neurons, provide the information that we use to navigate in the visual world. These responses largely originate from the excitatory and inhibitory interactions among bipolar, amacrine, and RGCs in the inner plexiform layer (IPL). The IPL is divided into 10 strata[6] where ~13 types of bipolar cells[2,7,8] make excitatory synapses with co-stratified RGCs. The selectivity of RGC responses is further refined by interneuron amacrine cells (ACs), numbering ~60 types[9–11], that make predominantly inhibitory synapses onto co-stratified bipolar cells and RGCs. While it is acknowledged that each type of AC is associated with a distinct functional role in visual processing, the specific roles have been elucidated for fewer than 20 AC types[12–24]. For most of the remaining ACs, we are constrained from assigning precise functions

due to a fundamental gap in our understanding of their functional properties and connectivity patterns.

RGCs with similar stratification patterns are positioned to receive similar bipolar and AC inputs but often respond differently to visual stimuli For instance, HD1-RGC, HD2-RGC, W3(UHD)-RGC, and LED RGC have dendrites that stratify in the middle of the inner plexiform layer (IPL), but they exhibit distinct response properties to object size, velocity, and foreground-background motion[25]. To distinguish moving objects within a visual scene that is often itself in motion, neurons need to sense when movement in the receptive field center differs from that in the surround[26]. The object motion sensitivity (OMS) in RGCs may originate from ACs[15,27,28] and the center-surround receptive fields of bipolar cells[29]. In the mouse retina, the W3(UHD)-RGC circuit is the most extensively studied OMS pathway. This circuit allows W3(UHD)-RGCs to respond to local motion but not to global image motion[15,25,27,30]. The object motion sensitivity of W3(UHD)-RGCs originates from two types of ACs: VGlut3-AC and TH2-AC. VGlut3-AC, a narrow-field AC with excitatory outputs[15,23,31], is itself OMS and adds,

[1]Departments of Ophthalmology and Neuroscience, Feinberg School of Medicine, Northwestern University, Chicago, IL 60611, USA. [2]These authors contributed equally: Andrew Jo, Sercan Deniz. ✉e-mail: yongling-zhu@northwestern.edu

with a slight delay to the excitatory drive from bipolar cells to W3 (UHD)-RGCs during local motion[15,27,32]. On the other hand, the TH2-AC, a wide-field AC, suppresses center bipolar cell inputs when the center and surround patterns move in synchrony, which contributes to the object motion sensitivity of W3(UHD)-RGCs[28]. Besides ACs, glutamate release from bipolar cells can enhance the saliency of objects entering the receptive field, thereby aiding in the detection of novel objects[29].

Here, we used a Cre/tTA intersectional strategy[33,34] to restrict AC labeling and achieve single-cell type labeling of a newly identified AC type, COMS-AC (counter-OMS AC). COMS-AC is a narrow-field AC that stratifies in the middle of the IPL and exhibits strong OMS. We found that COMS-AC provides inhibitory glycinergic input to HD2p-RGC (putative HD2-RGC), which lacks OMS, but not to the OMS W3 (UHD)-RGC. This inhibition suppresses HD2p-RGC's response during local motion stimuli, but not during global motion stimuli. Same as W3 (UHD)-RGC, HD2p-RGC receives suppression from TH2-AC during global motion stimuli. As a result of the dual local and global motion suppressive inputs, HD2p-RGC responds poorly to moving stimuli. Our results reveal an OMS-canceling AC pathway mediated by COMS-AC and show that HD2p-RGC and W3 (UHD)-RGC combine shared inhibition to global motion and dissimilar excitation/inhibition to local motion. Consequently, HD2p-RGC is insensitive to OMS, while W3 (UHD)-RGC is OMS-sensitive. The study shows how local AC circuits can produce distinct response properties in structurally similar, co-stratifying RGC types in the retina.

## Results

### COMS-AC identified in a Camk2a-tTA driver

The Cre/tTA intersectional strategy relies on two orthogonal binary systems, Cre/loxP and tTA/TRE[35]. The Camk2a-tTA driver expresses a tetracycline-controlled transactivator protein promoter[36]. To examine tTA-expression patterns in the retina, we crossed the Camk2a-tTA line with CMV-Cre (a ubiquitous Cre driver) and used a Cre/tTA dependent-GCaMP6f reporter line Ai93[34] to label cells in the intersection. In CMV-Cre;Camk2a-tTA;Ai93 mice, tTA was expressed in several AC types in the inner nuclear layer (INL), as well as many RGC types in the ganglion cell layer (GCL) (Fig. 1c). To specifically label ACs and remove the labeling from RGCs, we crossed Camk2a-tTA with VGAT-Cre which is driven by the Slc32a1 promoter (Fig. 1a, b). In VGAT-Cre;Camk2a-tTA;Ai93 mice, most labeling of RGCs was removed, leaving only two types of narrow-field ACs labeled (Fig. 1di–iii), along with 1–5 wide-field ACs labeled per retina. The narrow-field ACs in VGAT-Cre;Camk2a-tTA;Ai93 mice can be classified into two types based on their morphologies (Fig. 1dii–iii, Table S1). These two types occurred at a ratio ranging from 4:1 to 9:1 (COMS-AC:CK2-AC2). The majority type, named COMS-AC (Fig. 1dii), was bistratified with most of its dendrites ramifying in a broad band in the middle of the IPL ($43 \pm 9\%$ of the IPL, $n = 22$) and the rest in a narrow band close to the INL ($10 \pm 4\%$ of the IPL, $n = 22$) (Fig. 1f, g, Table S1). The minority type, named CK2-AC2, ramified in three bands: one narrow band adjacent to the OFF-ChAT band ($42 \pm 8\%$ of the IPL, $n = 15$), one narrow band overlapping with the ON-ChAT band ($65 \pm 5\%$ of the IPL, $n = 15$), and a diffuse band close to the GCL ($83 \pm 5\%$ of the IPL, $n = 15$) (Fig. 1h, i, Table S1). Both types were identified as glycinergic based on the presence of GlyT1 immunoreactivity (Fig. 1j). Morphological analysis suggested that COMS-AC resembled AC type 19, while CK2-AC2 resembled type 42 in a previous serial block-face electron microscopy (SBEM) study[11]. Additionally, COMS-AC showed a strong resemblance to nGnG1[9,37] and expressed PPP1R17 (Fig. 1k, Supplementary Fig. 1c, d) which is also expressed in nGnG1 and nGnG3[9]. We speculate that COMS-AC corresponds to nGnG1 (cluster 24 in single-cell RNA sequencing[9]), while "Müller glia-coupled amacrine cell"[20] corresponds to nGnG3. Both COMS-AC and CK2-AC2 displayed a distribution that was not biased towards any specific quadrant (Supplementary Fig. 1a). It is worth noting that our intersectional strategy only labeled subpopulations of each cell type,

and further confirmation of their distribution patterns will be necessary through future studies.

We focused our functional analysis on COMS-AC, as it represents the majority of ACs labeled in Camk2a-tTA mice. To isolate COMS-ACs from CK2-AC2s and other AC types, we replaced VGAT-Cre with an inducible VGAT-iCreER driver (Fig. 1e). This allowed us to label COMS-AC more specifically by inducing Cre expression at certain time points. Tamoxifen injection after P100 removed labeling from CK2-AC2s and wide-field ACs, leaving COMS-AC as the only cell type labeled (Fig. 1ei-iv). The density recovery profiles (DRPs) showed an exclusion zone between labeled ACs in VGAT-iCreER;Camk2a-tTA;Ai93 mice (Fig. 1ev), but not in VGAT-Cre;Camk2a-tTA;Ai93 mice (Fig. 1div). The quantification of labeled cells stratification revealed two peaks ($7.5 \pm 0.2\%$ and $41.7 \pm 0.1\%$), aligning with the exclusive labeling pattern of COMS-AC cells (Supplementary Fig. 1b). Meanwhile, all the labeled ACs exhibited positive PPP1R17 expression although only 23.8% of PPP1R17-positive cells were labeled in VGAT-iCreER;Camk2a-tTA;Ai93 mice (Supplementary Fig. 1c, d). These results indicate that the labeling of CK2-AC2 and other ACs was successfully removed.

### COMS-AC responds to light decrements

Given that COMS-AC dendrites primarily ramify in the middle of the IPL spanning the boundary between ON and OFF layers, we investigated how visual responses were organized across the laminar depth of COMS-AC dendrites. To address this, we performed GCaMP6f imaging in COMS-AC in VGAT-Cre;Camk2a-tTA;Ai93 mice while applying a spot of light (50 μm diameter) in the receptive field center. Using two-photon imaging, we recorded GCaMP6f responses at three layers: the soma, the proximal dendrites stratifying in the OFF layer, and the distal dendrites stratifying in the middle of the IPL (Fig. 2a). COMS-AC elicited a transient OFF response across all three layers (Fig. 2b). The OFF response was blocked by UBP310 which blocks the transmission from photoreceptors to OFF bipolar cells, suggesting that COMS-AC cells receive excitatory inputs from OFF bipolar cells (Fig. 2c). We conducted two experiments to assess the receptive fields of COMS-AC. One experiment involved an expanding light spot on a dark background (Fig. 2d), while the other used an expanding dark spot on a gray background (Fig. 2f). Our results revealed that both proximal and distal dendrites exhibited similar receptive fields, characterized by peaks at -60 μm in response to the offset of the light (Fig. 2e) and 75 μm in response to light decrements (Fig. 2g). To confirm that COMS-AC is depolarized in the dark, we recorded changes in membrane potential in response to light stimuli. As depicted in Fig. 2h, i, COMS-AC had a resting potential of $-52.1 \pm 2.0$ mV ($n = 9$). When a 75 μm diameter light spot was applied to the receptive field center, COMS-AC showed a sustained small hyperpolarization to $-58 \pm 2.0$ mV ($n = 9$) in the light, followed by a transient depolarization to $-26.6 \pm 2.4$ mV ($n = 9$) at the offset of the light. Therefore, both GCaMP6f imaging and whole-cell recording demonstrated that COMS-AC received excitation from OFF bipolar cells and acted as a transient OFF AC. The ON hyperpolarization in Fig. 2h could result from a reduction of OFF bipolar cell excitatory input in the light and/or increased inhibition from other ACs.

To investigate the contribution of other ACs to the receptive field of COMS-AC, we applied gabazine (a selective $GABA_A$ antagonist), TPMPA (a selective $GABA_C$ antagonist), strychnine (a selective glycine receptor antagonist), or MFA (a gap junction blocker) in the bath. We observed significant effects on GCaMP6f responses with TPMPA (Fig. 3a), gabazine (Fig. 3b), and MFA (Fig. 3d). TPMPA demonstrated the most pronounced effects, increasing GCaMP6f responses in all tested spots, particularly in large spots. At 150 μm, TPMPA resulted in a 6-fold increase in GCaMP6f responses, whereas gabazine led to a 2.5-fold increase. MFA reduced GCaMP6f responses by 14% at 75 μm. In contrast, strychnine (Fig. 3c) had minimal to no effect. These results suggest that most of the inhibition to COMS-AC arises from GABAergic ACs and is primarily mediated by $GABA_C$ receptors, although $GABA_A$

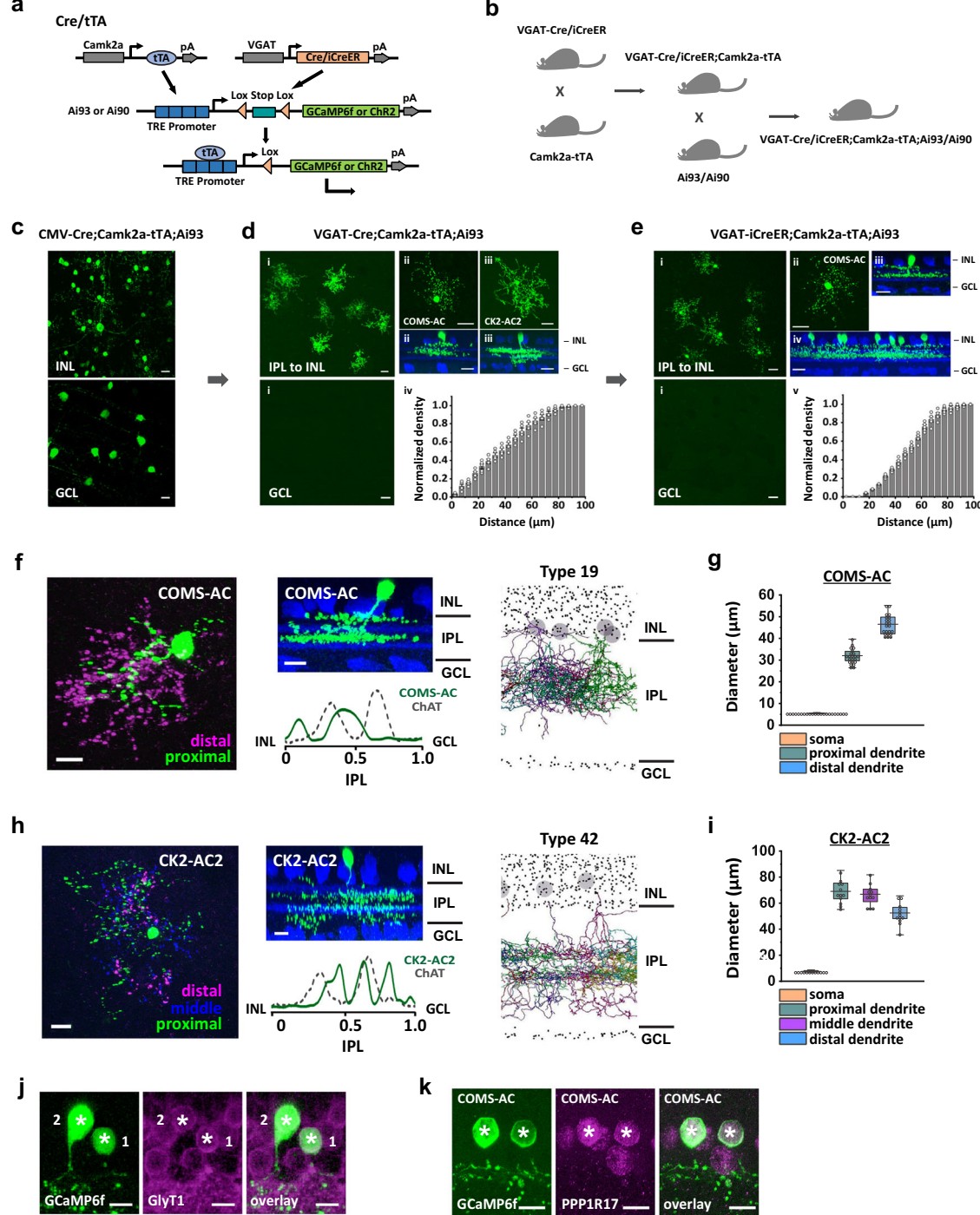

**Fig. 1 | COMS-AC and CK2-AC2 identified with Cre/tTA intersectional strategy.**
**a** VGAT-Cre(iCreER)/Camk2a-tTA intersectional strategy. Ai93 (or Ai90) is a Cre/tTA dependent-GCaMP6f (or Chronos) reporter line. **b** The triple transgenic breeding scheme for labeling ACs in Camk2a-tTA driver. **c** CMV-Cre;Camk2a-tTA;Ai93 labeled multiple AC types and RGCs. Experiments were replicated independently in at least 30 retinae with similar results. **d** In VGAT-Cre;Camk2a-tTA;Ai93, labeling was removed from RGCs, leaving only COMS-AC, CK2-AC2, and a few wide-field ACs. (I) The distribution of GCaMP6f-expressing cells in the INL, IPL, and GCL is shown. (ii, iii) The flat-mount view (top) and side view (bottom, with ChAT in blue) of COMS-AC (ii) and CK2-AC2 (iii) are presented. (iv) The density recovery profiles (DRPs) of GCaMP6f-expressing cells in the INL are shown, with $n = 5$ retinae.
**e** Replacing VGAT-Cre with VGAT-iCreER and injecting tamoxifen after P100 led to exclusive labeling of COMS-AC. (i) The distribution of GCaMP6f-expressing cells in the INL, IPL, and GCL is shown. The flat-mount view (ii) and side view (iii) of COMS-AC labeled in the intersection are presented. (iv) The side view of a large retinal

region displaying the uniform dendritic stratification of cells in the intersection is presented, with ChAT staining in blue. (v) The density recovery profiles (DRPs) of GCaMP6f-expressing cells in the INL are shown, with $n = 6$ retinae. Morphologies of COMS-AC (**f**) and CK2-AC2 (**h**). The flat-mount views are on the left, the side views with ChAT in blue are in the middle, and the corresponding cell types in SBEM[11] are on the right. Experiments were replicated independently in at least 40 cells with similar results. Soma and dendritic diameters of COMS-AC (**g**) and CK2-AC2 (**i**) are shown, with $n = 22$ cells for COMS-AC and $n = 15$ cells for CK2-AC2.
**j** Immunostaining with GlyT1 antibody showed that both COMS-AC (1) and CK2-AC2 (2) were glycinergic. **k** COMS-AC was positive to immunostaining of PPP1R17. Scale bars: 20μm (**c**, **d**, and **e**), 10 μm (**f**, **h**, **j**, and **k**). **j**, **k** Experiments were replicated independently in at least 20 cells with similar results. **d**$_{iv}$ and **e**$_{v}$ The data are presented as mean ± SEM. The box plots display the mean, 25th, and 75th percentiles, while the whiskers indicate the 1.5 interquartile range. Source data of (**g**) and (**i**) are provided as a Source Data file.

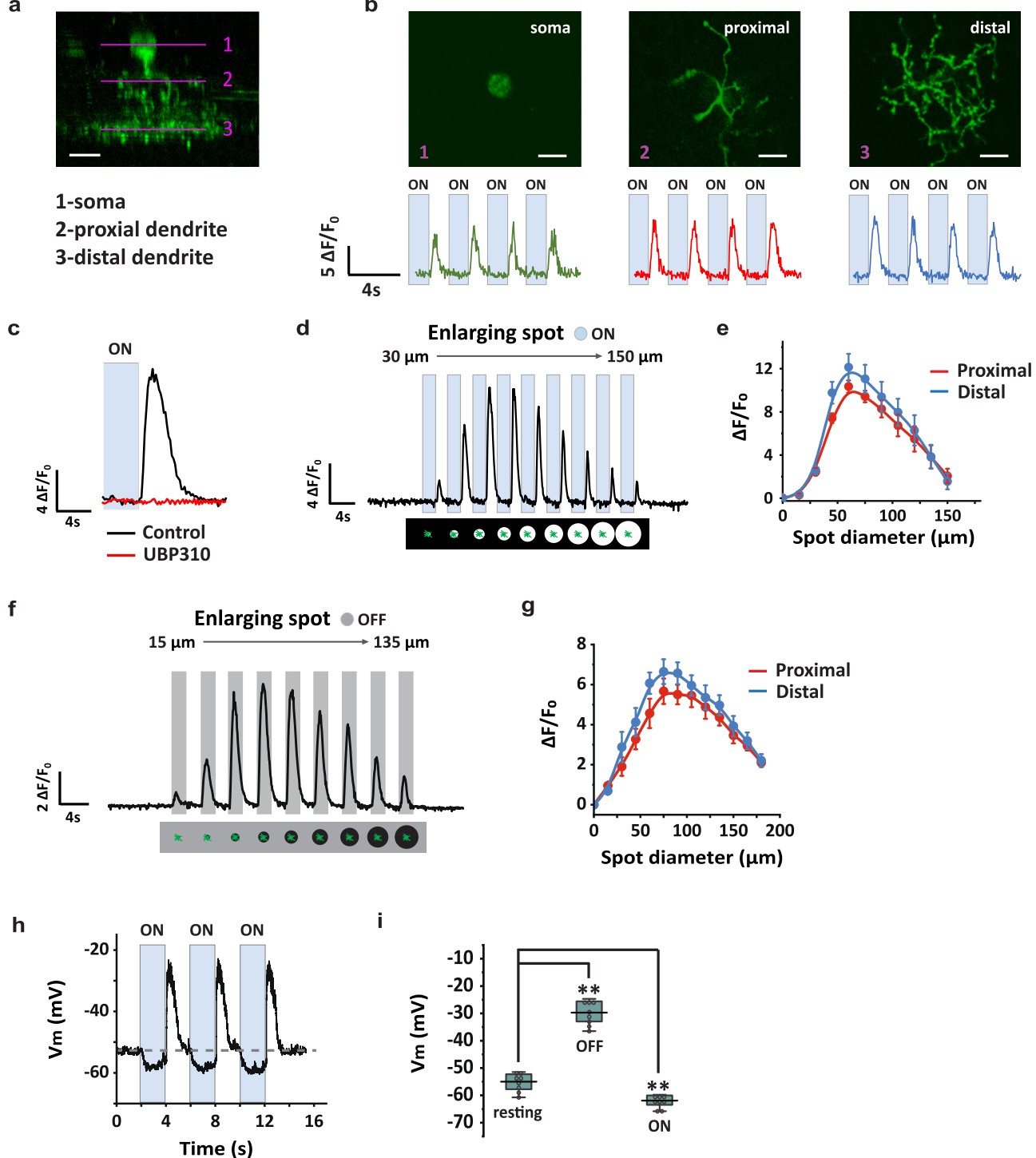

**Fig. 2 | Light responses of COMS-AC. a** COMS-AC was labeled with GCaMP6f, and the soma (1), proximal dendrites (2), and distal dendrites (3) were imaged at three different focal planes, with a scale bar of 10 μm. Experiments were replicated independently in at least 40 cells with similar results. **b** The two-photon images at each focal plane in (**a**) are displayed at the top, with a scale bar of 10 μm. The GCaMP6f signals of each focal plane in response to a 50 μm spot (100% contrast, blue bar) flashing over the receptive field center are presented at the bottom. **c** Blockade of responses in the distal dendrites by UBP310 (25 μM). Experiments were replicated independently in at least 20 cells with similar results. The receptive field was probed by applying an expanding light spot (100% contrast, blue bar) against a dark background (0% contrast) (**d**). The normalized peak responses in the distal and proximal dendrites are summarized in (**e**). N = 8 cells, error bars represent SEM. The receptive field was probed by applying an expanding dark spot (0% contrast, gray bar) against a gray background (50% contrast) (**f**). The normalized peak responses in the distal and proximal dendrites are summarized in (**g**), with n = 7 cells, and error bars representing SEM. **h** COMS-AC was targeted for whole-cell recording. The membrane potential was hyperpolarized in the light and depolarized in the dark in response to a 50 μm flashing spot (100% contrast, blue bar). **i** Summarized membrane potential at rest, in the dark (OFF), and in the light (ON), n = 8 cells. ON: **$p$ = 0.0078, Wilcoxon Signed Rank test, two tailed. OFF: **$p$ = 0.0078, Wilcoxon Signed Rank test, two tailed. The box plot displays the mean, 25th, and 75th percentiles, while the whiskers indicate the 1.5 interquartile range. **b, c** Experiments were replicated independently in at least 20 cells with similar results. Source data of (**e, g**), and (**i**) are provided as a Source Data file.

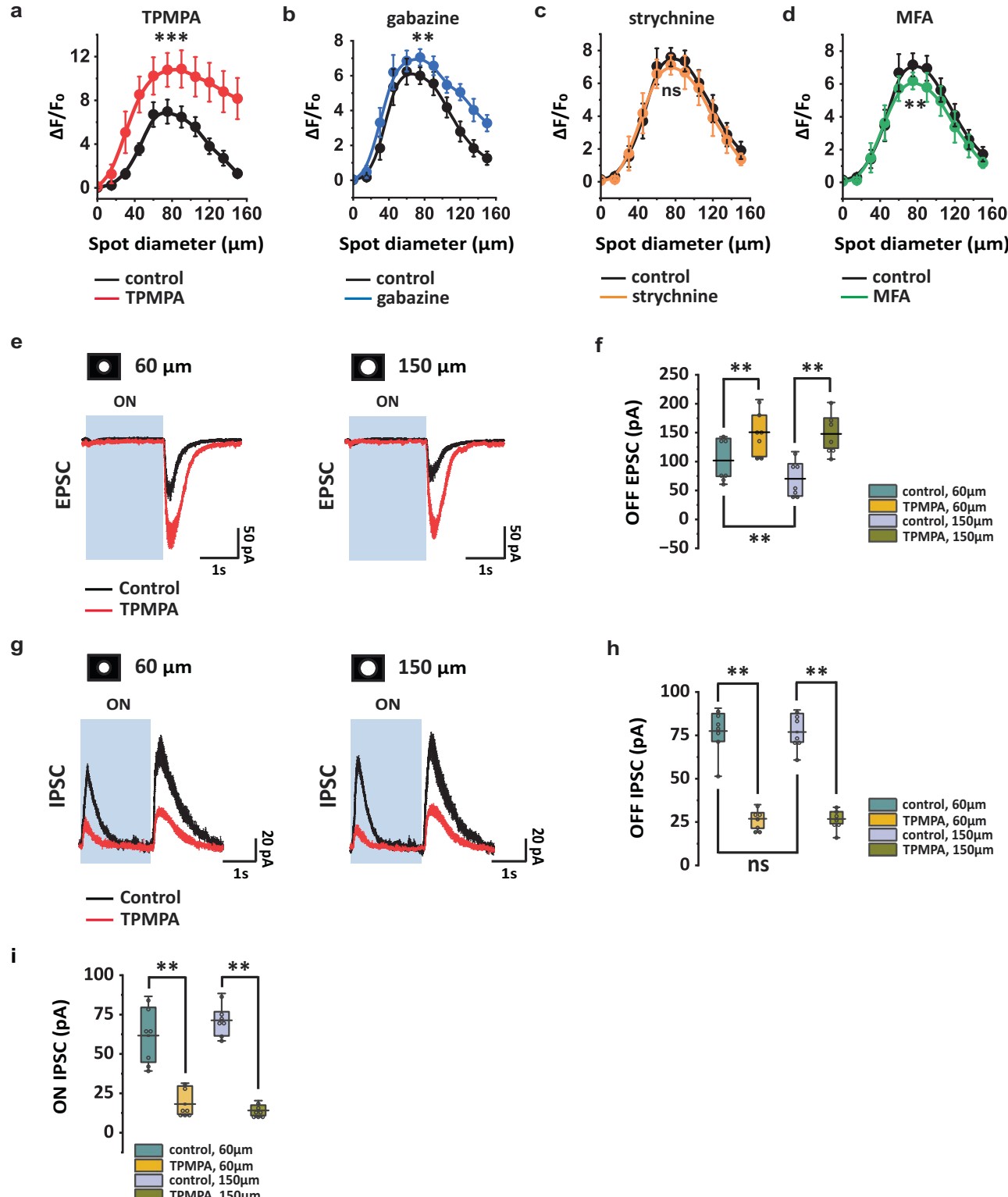

**Fig. 3 | Receptive field properties of COMS-AC. a-d** Effects of TPMPA (**a**), gabazine (**b**), strychnine (**c**), and MFA (**d**) on the receptive field of COMS-AC, measured with GCaMP6f responses. Error bars represent SEM. **a** n = 10 cells, ***p = 0.00094. **b** n = 7 cells, **p = 0.0020. **c** n = 6 cells, ns: p = 0.054. **d** n = 6, **p = 0.0068. Wilcoxon Signed Rank test, two tailed. **e** TPMPA increased OFF EPSCs in response to both 60 μm and 150 μm light spots. **f** Summarized OFF EPSCs in control and TPMPA. N = 7 cells, **p = 0.0078 for all, Wilcoxon Signed Rank test, one tailed. **g** TPMPA reduced both ON and OFF IPSCs in response to 60 μm and 150 μm light spots. **h, i** Summarized OFF IPSCs (**h**) and ON IPSCs (**i**) in control and TPMPA. N = 7 cells. **h** **p = 0.0078 for both 60 μm and 150 μm, Wilcoxon Signed Rank test, one tailed. ns: p = 0.81, Wilcoxon Signed Rank test, two tailed. **i** **p = 0.0078 for both 60 μm and 150 μm, Wilcoxon Signed Rank test, one tailed. **f, h,** and **i** The box plots display the mean, 25th, and 75th percentiles, while the whiskers indicate the 1.5 interquartile range. Source data of **a-d, f, h,** and **i** are provided as a Source Data file.

receptors also contribute to the inhibition. Moreover, electrical coupling also plays a role in shaping the receptive field of COMS-AC.

To determine the location of GABA$_C$-mediated inhibition, we measured EPSCs and IPSCs of COMS-AC in control and in the presence of TPMPA. In control conditions, COMS-AC responded to the offset of the light with both an EPSC (Fig. 3e) and an IPSC (Fig. 3g), while only an IPSC was observed at the onset of the light without an EPSC (Fig. 3e, g). This absence of an ON EPSC was consistent with the lack of an ON response in GCaMP6f recordings (Fig. 2b), while the presence of an ON IPSC may contribute to the ON hyperpolarization in current-clamp recordings (Fig. 2h). When 50 μM TPMPA was applied to the bath, the OFF EPSC increased (Fig. 3e, f), while both the OFF IPSC and ON IPSC (Fig. 3g, i) decreased, indicating that GABA$_C$-mediated inhibition occurs both presynaptically at bipolar cell terminals and postsynaptically at COMS-AC neurites.

## COMS-AC is object motion sensitive

We investigated whether COMS-AC encodes specific image features. Since COMS-AC dendrites primarily ramify in the middle of the IPL where several object motion-sensitive RGCs stratify, such as W3 (UHD)-RGCs and HD1-RGCs, we examined whether COMS-AC responds differently to global and local motion stimuli. Interestingly, we found that COMS-AC responds strongly to local motion, but not to global motion in both distal dendrites and proximal dendrites (Fig. 4a, b), with a local motion preference index (LMPI) of $0.94 \pm 0.01$ for distal dendrites ($n = 22$) and $0.90 \pm 0.03$ for proximal dendrites ($n = 7$) (Fig. 4c). These results suggest that COMS-AC possesses the property of object motion sensitivity (OMS), a feature shared by another AC type, the VGlut3-AC.

We next performed whole-cell recording to measure the changes in membrane potential of COMS-AC in response to global and local motion stimuli. Consistent with the GCaMP6f recording, global motion elicited an overall hyperpolarization in COMS-AC with a small transient depolarization ($0.35 \pm 0.09$ s) from $-57.5 \pm 1.1$ mV to $-53.5 \pm 1.9$ mV, followed by a larger sustained hyperpolarization ($4.75 \pm 0.30$ s) to $-67.5 \pm 1.0$ mV ($n = 7$) (Fig. 4d, left). Conversely, local motion stimulation evoked a more sustained ($1.57 \pm 0.03$ s) and larger depolarization from $-56.8 \pm 1.6$ mV to $-43.6 \pm 2.5$ mV, followed by little hyperpolarization (-2 mV) ($n = 7$) (Fig. 4d, right). We then compared the EPSCs and IPSCs evoked by global and local motion. Consistent with the changes in membrane potential, global motion produced smaller EPSCs ($35.0 \pm 8.6$ pA vs. $116.9 \pm 15.2$ pA, $n = 7$) and larger IPSCs ($83.4 \pm 7.5$ pA vs. $35.6 \pm 3.5$ pA, $n = 7$) than local motion (Fig. 4e, f). These results suggest that during global motion, COMS-AC receives strong presynaptic and postsynaptic inhibition that together produce an overall hyperpolarization.

We investigated the origin of the OMS in COMS-AC by examining the contribution of other ACs through the application of gabazine, TPMPA, strychnine, or MFA in the bath. We found that TPMPA produced a significant increase in COMS-AC responses to global motion, as measured by GCaMP6f recording, without affecting local motion (Fig. 4g, h). As a result, the LMPI was reduced from $0.94 \pm 0.01$ to $0.18 \pm 0.07$ ($n = 10$, Fig. 4i). Whole-cell recording revealed a three-fold increase in EPSC from $35.0 \pm 8.6$ pA to $118.2 \pm 15.0$ pA ($n = 7$) and a 50% decrease in IPSC from $83.4 \pm 7.5$ pA to $37.5 \pm 7.2$ pA ($n = 7$) in response to global motion (Fig. 4j, k), without affecting local motion (Fig. 4l, m). The EPSC ($118.2 \pm 15.0$ pA) and IPSC ($37.5 \pm 7.2$ pA) during global motion were similar to those during local motion ($116.9 \pm 15.2$ pA for EPSC, $35.6 \pm 3.5$ pA for IPSC) in control. The results indicated that the differences in COMS-AC responses to global motion and local motion were primarily due to GABA$_C$-mediated inhibition that was activated during global motion. In support of this notion, gabazine, strychnine, and MFA had no effect on either local motion or global motion (Supplementary Fig. 2). These findings suggest that GABA$_C$-mediated presynaptic and postsynaptic inhibition in global motion contribute to the OMS in COMS-AC (Fig. 4n).

TH2-AC co-stratifies with COMS-AC and demonstrates varying kinetics in response to global and local motion[28]. It produces rapid responses to global motion to counteract excitatory input in W3 (UHD)-RGC but exhibits slower responses to local motion that do not activate GABA release and peak after excitation in W3 (UHD)-RGC. We speculated whether the OMS observed in COMS-AC might arise from TH2-AC through a similar mechanism as that observed in W3 (UHD)-RGC. To investigate this possibility, we created a quintuple mouse line, VGAT-Cre;Camk2a-tTA;Ai93;TH-2A-FlpO;RC::FPDi, where TH-2A-FlpO is a Flp recombinase line regulated by the TH (tyrosine hydroxylase) promoter[38] and RC::FPDi is a Cre/Flp-dependent hM4Di silencing line[39]. In this quintuple mouse line, hM4Di[40] was expressed in TH2-AC using the VGAT-Cre;TH-2A-FlpO;RC::FPDI combination, whereas GCaMP6f was expressed in COMS-AC with the

VGAT-Cre;Camk2a-tTA;Ai93 combination (Supplementary Fig. 3a). We found that silencing TH2-ACs with the DREADD agonist Compound 21 (C21, 10 μM)[41] in the bath had no discernible effect on either the receptive field (Supplementary Fig. 3b, c) or the OMS (Supplementary Fig. 3d, e) of COMS-AC. These findings strongly suggest that TH2-AC is not the source of the GABA$_C$-mediated inhibition that shapes the receptive field and OMS in COMS-AC. Therefore, the OMS observed in COMS-AC is likely to originate from other ACs.

## COMS-AC provides glycinergic input to HD2p-RGC but not to W3 (UHD)-RGC

The strong OMS of COMS-AC suggests that it may play a role in object motion detection in its postsynaptic RGCs. Our next step was to map its postsynaptic RGCs with optogenetic activation. COMS-AC primarily stratifies in the middle of the IPL; therefore, we focused on two RGC types, W3-RGC (also known as UHD-RGC, an OMS sensitive RGC) and HD2-RGC (an OMS insensitive RGC), that co-stratify with COMS-AC. To selectively express ChR2 in COMS-AC while labeling W3 (UHD)-RGC and HD2-RGC with fluorescent proteins for targeted whole-cell recording (Fig. 5b), we created a quadruple mouse line (VGAT-iCreER;Camk2a-tTA;Ai90;W3-YFP) (Fig. 5a). Ai90 is a Cre/tTA-dependent reporter line that expresses Chronos[34,42], a more sensitive and faster version of ChR2. W3-YFP[43] is a YFP line that labels W3 (UHD)-RGC, HD2-RGC, and several other RGC types (Gregory W. Schwartz, personal communication).

We first patched on Ai90 labeled COMS-AC to determine the range of optogenetic stimulus strengths (Fig. 5c) that give similar levels of depolarizations to those produced by photoreceptor stimulation (~25 mV, Fig. 2h, i), as well as to investigate the kinetics of ChR2-mediated depolarization (Supplementary Fig. 4). The depolarization of COMS-AC started at $5.3 \pm 0.2$ ms after the onset of the blue light, exhibiting a rise time of $1.8 \pm 0.1$ ms from 20% to 80% of the peak amplitudes. To identify HD2-RGC and W3 (UHD)-RGC, we performed cell-attached recording on YFP-labeled RGCs and measured their responses to light to evaluate their spiking patterns (Fig. 5e, k), receptive fields (Fig. 5f, l), and OMS (Fig. 5g, m). HD2-RGC is OFF-dominant and OMS insensitive[25], while W3 (UHD)-RGC showed a balanced ON-OFF response and a strong OMS[15,25,27,30]. Once we confirmed the identity of the targeted RGC, we conducted whole-cell recording and clamped the membrane potential to 0 mV. Next, we added a mixture of blockers (L-AP4 + CNQX + UBP310 + D-APV) to the bath to inhibit glutamatergic transmission from photoreceptors to bipolar cells. We discovered that optogenetic stimulation with blue light induced robust IPSCs in all 25 RGCs (Fig. 5h) that closely resembled HD2-RGCs in terms of morphology, spiking patterns, receptive fields, and OMS responses. These RGCs have been designated as HD2p-RGCs (putative HD2-RGC) instead of HD2-RGC due to the different stimulus we employed for OMS assessment in comparison to the original study[25]. The IPSC started to rise ~7 ms after the onset of ChR2 activation, with a synaptic latency of ~2 ms following the onset of COMS-AC depolarization (Supplementary Fig. 4). The IPSCs were eliminated by strychnine (Fig. 5h), indicating that

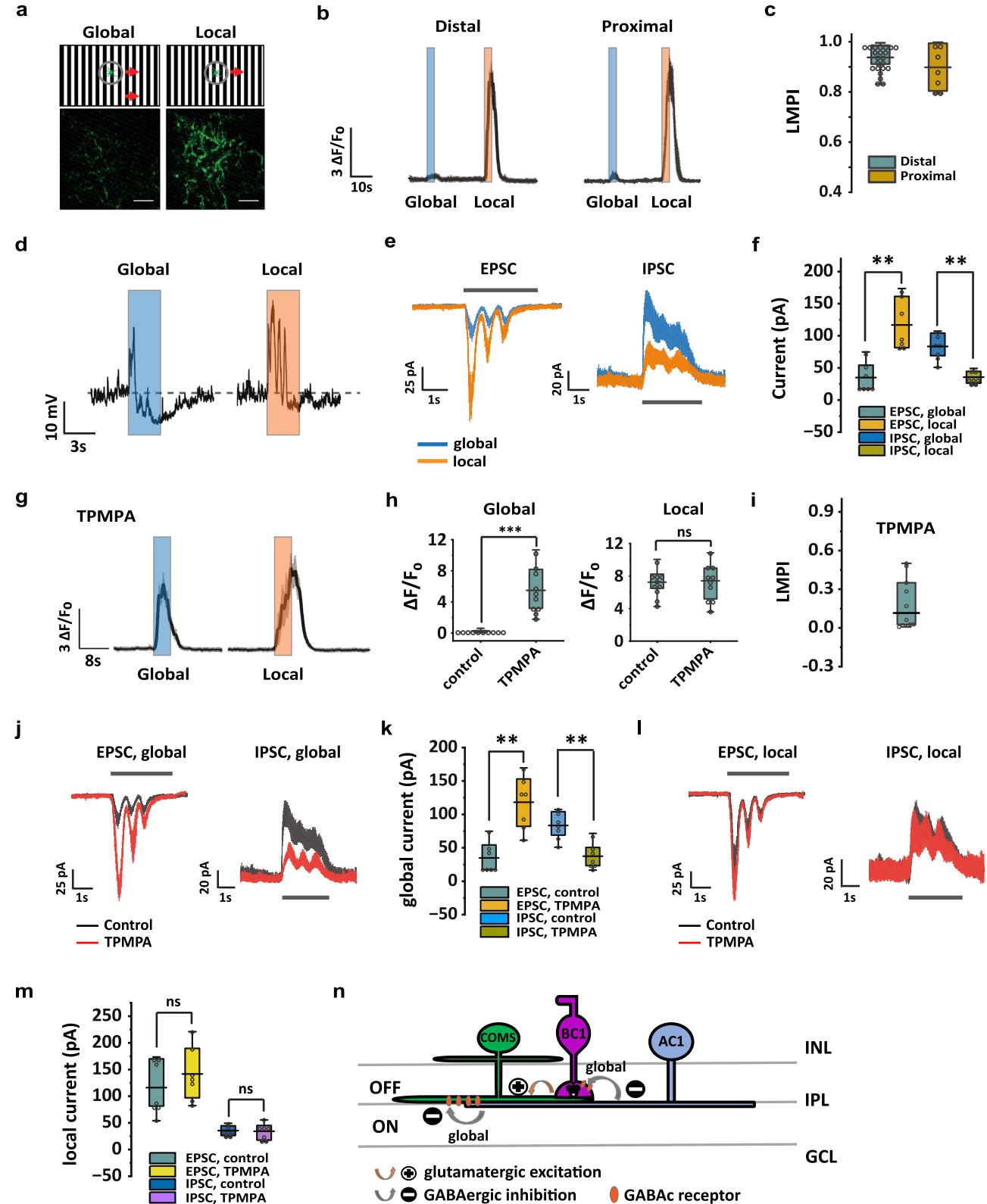

COMS-AC offers glycinergic input to HD2p-RGCs. However, because the distribution of ChR2-labeled COMS-ACs throughout the retina may not be uniform owing to tamoxifen-induced Cre expression, we opted to normalize IPSC amplitude concerning an individual COMS-AC. Following the recording, we fixed the retina, confirmed RGC identity via morphology (Fig. 5d, j), and counted the number of presynaptic COMS-ACs. The number of presynaptic COMS-ACs varied from 5 to 14 for each HD2p-RGC, with an average of 9.1 ± 0.6 ($n = 25$). We estimated that each

COMS-AC produced an IPSC of 13.0 ± 0.9 pA ($n = 25$) in an HD2p-RGC (Fig. 5i). Conversely, optogenetic stimulation of COMS-AC evoked either no IPSCs (5/12 RGCs) or small IPSCs (≤3 pA from each COMS-AC, 7/12 RGCs) in W3 (UHD)-RGC, resulting in an average IPSC of 1.3 ± 0.3 pA ($n = 12$) from each COMS-AC (Fig. 5n). As a result, we concluded that even though COMS-AC co-stratifies with both HD2p-RGC and W3 (UHD)-RGC, it selectively establishes synaptic connections with HD2p-RGC, not with W3 (UHD)-RGC.

**Fig. 4 | COMS-AC is object motion sensitive. a** COMS-AC preferred local motion over global motion. At the top, a schematic illustrating the motion stimulus is presented. At the bottom, the GCaMP6f response in the distal dendrites is shown to be weak in global motion (left) and strong in local motion (right), with a scale bar of 10 μm. Experiments were replicated independently in at least 30 cells with similar results. **b** GCaMP6f responses recorded in the distal dendrites and proximal dendrites during global motion and local motion, $n = 22$ for the distal dendrites, $n = 7$ for the proximal dendrites. **c** Local motion preference index (LMPI) of COMS-AC, $n = 22$ for the distal dendrites, $n = 7$ for the proximal dendrites. LMPI = $(\Delta F/F_{0, local} - \Delta F/F_{0, global})/(\Delta F/F_{0, local} + \Delta F/F_{0, global})$. **d** Whole-cell recording of membrane potential in response to global motion stimulus and local motion stimulus. **e** EPSC and IPSC in response to global motion stimulus and local motion stimulus. **f** COMS-AC showed weaker EPSC and stronger IPSC in global motion than local motion. $N = 7$ cells, **$p = 0.0078$ for both, Wilcoxon Signed Rank test, one tailed. **g** GCaMP6f responses in TPMPA, recorded in the distal dendrites, $n = 10$ cells. **h** TPMPA increased GCaMP6f response in global motion, but not in local motion. $N = 10$ cells, ***$p = 0.00098$, Wilcoxon Signed Rank test, one tailed. ns: $p = 0.70$, Wilcoxon Signed Rank test, two tailed. **i** LMPI of COMS-AC in TPMPA, $n = 10$ cells. **j, k** TPMPA increased EPSC and decreased IPSC in global motion. $N = 7$ cells, **$p = 0.0078$ for both, Wilcoxon Signed Rank test, one tailed. **l, m** TPMPA had no effect on EPSC and IPSC in local motion. $N = 7$ cells, ns: $p = 0.11$ for EPSC, $p = 0.94$ for IPSC, Wilcoxon Signed Rank test, one tailed. **n** Model illustrating the Object Motion Sensitivity (OMS) of COMS-AC. COMS-AC receives GABAc-mediated inhibition from an AC (AC1) in global motion, but not in local motion, through both presynaptic and postsynaptic mechanisms. **c, f, h, i, k** and **m** The box plots display the mean, 25th, and 75th percentiles, while the whiskers indicate the 1.5 interquartile range. Source data of (**c, f, h, i, k**, and **m**) are provided as a Source Data file.

## Generation and validation of TITL-PSAM⁴-GlyR-GCaMP6s mouse line

To investigate how COMS-AC shapes the responses of HD2p-RGCs, we developed a Cre/tTA-dependent silencing mouse line called TITL-PSAM⁴-GlyR-GCaMP6s, where both PSAM⁴-GlyR[44] and GCaMP6s[45] are co-expressed in the presence of Cre and tTA (Fig. 6a). We positioned GCaMP6s behind PSAM⁴-GlyR to enable direct reporting of inhibition caused by PSEM, the agonist for PSAM⁴-GlyR. To validate the efficacy of TITL-PSAM⁴-GlyR-GCaMP6s in COMS-AC, we created triple transgenic mice (VGAT-iCreER;Camk2a-tTA;TITL-PSAM⁴-GlyR-GCaMP6s), as shown in Fig. 6b. In COMS-ACs that express both PSAM⁴-GlyR and GCaMP6s (Fig. 6c), uPSEM⁷⁹², an ultrapotent PSEM agonist for PSAM⁴-GlyR[44], effectively inhibited GCaMP6s responses in a dose-dependent manner (Fig. 6d, e). In mice homozygous for PSAM⁴-GlyR, uPSEM⁷⁹² at a concentration of 10 nM blocked 96.6% ± 0.9% ($n = 10$) of GCaMP6s responses. In contrast, no change in GCaMP6f responses was observed in control COMS-AC, which only expressed GCaMP6f but not PSAM⁴-GlyR (Fig. 6f, g). These findings indicate that TITL-PSAM⁴-GlyR-GCaMP6s mice were effective in silencing COMS-ACs. We also evaluated the GCaMP6s signals generated by TITL-PSAM⁴-GlyR-GCaMP6s in COMS-AC. As illustrated in Supplementary Fig. 5, in comparison to Ai93 (TITL-GCaMP6f), TITL-PSAM4-GlyR-GCaMP6s produced roughly 20% of both the peak and baseline fluorescence levels, while demonstrating an equivalent $\Delta F/F_0$ response under the same stimulation.

## COMS-AC inhibits HD2p-RGC during local motion

To introduce PSAM⁴-GlyR into COMS-AC and label HD2p-RGC with YFP, we generated a quadruple mouse line: VGAT-iCreER;Camk2a-tTA;TITL-PSAM⁴-GlyR-GCaMP6s;W3-YFP (Fig. 7a). As demonstrated in Fig. 7b, c, exposure to 10 nM uPSEM⁷⁹² increased HD2p-RGC spike rates in response to the offset of light spots. The most significant effects of uPSEM⁷⁹² were observed in spots less than 100 μm, with a peak around 60 μm in diameter (Supplementary Fig. 6a). The spatial profile of uPSEM⁷⁹² effect was consistent with the receptive field of COMS-AC (Fig. 2e, g). These findings suggest that COMS-AC contributes to the center inhibition of the receptive field of HD2p-RGC. We investigated whether COMS-AC contributes to the ability of HD2p-RGC to detect object motion by examining HD2p-RGC spike rates in response to global and local motion in both control and 10 nM uPSEM⁷⁹². In control, HD2p-RGC exhibited minimal response to either global or local motion (Fig. 7d, upper panel). However, exposure to 10 nM uPSEM⁷⁹² increased the spike rate during local motion by approximately three-fold, with no significant effect on global motion (Fig. 7d, lower panel and Fig. 7e). Consequently, the LMPI increased from $0.29 \pm 0.03$ ($n = 7$) in control to $0.80 \pm 0.03$ ($n = 7$) in uPSEM⁷⁹² (Fig. 7f). These findings suggest that the inactivation of COMS-AC converts HD2p-AC from an OMS insensitive RGC to an OMS sensitive RGC. In contrast, the inactivation of COMS-AC had no impact on the OMS of W3 (UHD)-RGC (Supplementary

Fig. 7), which is consistent with the absence of glycinergic input from COMS-AC to W3 (UHD)-RGC (Fig. 5n).

To investigate the mechanism underlying the change in spike rates of HD2p-RGC, we examined the effects of uPSEM⁷⁹² on membrane potential, EPSCs, and IPSCs. We observed that uPSEM⁷⁹² did not affect the membrane potential in response to global motion, as shown in Fig. 7g, h. However, in response to local motion, uPSEM⁷⁹² induced a depolarization from $-51.4 \pm 1.8$ mV to $-41.5 \pm 2.4$ mV ($n = 5$), which triggered spike generation (Fig. 7g, i). The depolarization observed in local motion was caused by changes in both EPSC and IPSC. In control, the HD2p-RGC exhibited relatively large IPSCs and small EPSCs during both global motion (IPSC $218.1 \pm 30.6$ pA, EPSC $141.2 \pm 9.4$ pA, $n = 10$) and local motion (IPSC $277.6 \pm 28.9$ pA, EPSC $195.2 \pm 15.4$ pA, $n = 10$) (Fig. 7j–l). When exposed to 10 nM uPSEM⁷⁹², the EPSC and IPSC in global motion remained unchanged (Fig. 7j, k). In contrast, during local motion, the EPSC increased by 30.2% from $195.2 \pm 15.4$ pA to $254.1 \pm 21.3$ pA, while the IPSC decreased by 52.1% from $277.6 \pm 28.9$ pA to $132.9 \pm 17.3$ pA ($n = 10$) (Fig. 7j, l). It is worth noting that the effects of PSAM⁴-GlyR activation may have been underestimated since our protocol of tamoxifen induction combined with the intersectional strategy was unlikely to produce the expression of Cre and PSAM⁴-GlyR in the entire population of COMS-ACs. Nonetheless, these results indicate that COMS-AC provides both presynaptic and postsynaptic inhibition to HD2p-RGC during local motion, but not during global motion.

## TH2-AC provides GABAergic input to HD2p-RGC

HD2p-RGC exhibits poor responses to both global and local motion (Figs. 5g, 7d), indicating strong inhibition during both stimuli. While COMS-AC is a candidate for providing local inhibition, the source of global inhibition is unclear. TH2-AC is a possible candidate, as it co-stratifies with HD2p-RGC (Fig. 8d) and produces fast depolarizations to global motion but slow depolarizations to local motion[28]. To examine whether TH2-AC makes synaptic connections with HD2p-RGC, we expressed ChR2 in TH2-AC while labeling HD2p-RGC with YFP in TH-Cre;Ai32;W3-YFP mice (Fig. 8a). Our results (Fig. 8b, c) show that optogenetic stimuli with an intensity of 0.16 mW/mm² produced a depolarization of ~22 mV in TH2-AC, which is similar to the depolarization produced by photoreceptor stimulation (15 mV–26 mV)[28]. Optogenetic activation of TH2-AC at an intensity of 0.16 mW/mm² resulted in the production of IPSCs in all of the tested HD2p-RGCs (12/12), and these IPSCs could be fully blocked by a combination of gabazine and TPMPA (Fig. 8e, f). These findings suggest that TH2-AC provides GABAergic input to HD2p-RGC, similar to what has been observed in W3 (UHD)-RGC[28].

## TH2-AC inhibits HD2-RGC during global motion

In the TH-Cre mouse line, Cre expression was observed in TH2-ACs, as well as a small subset of dopaminergic ACs (DACs) and starburst ACs (SACs)[28]. To achieve a more precise labeling of TH2-ACs for

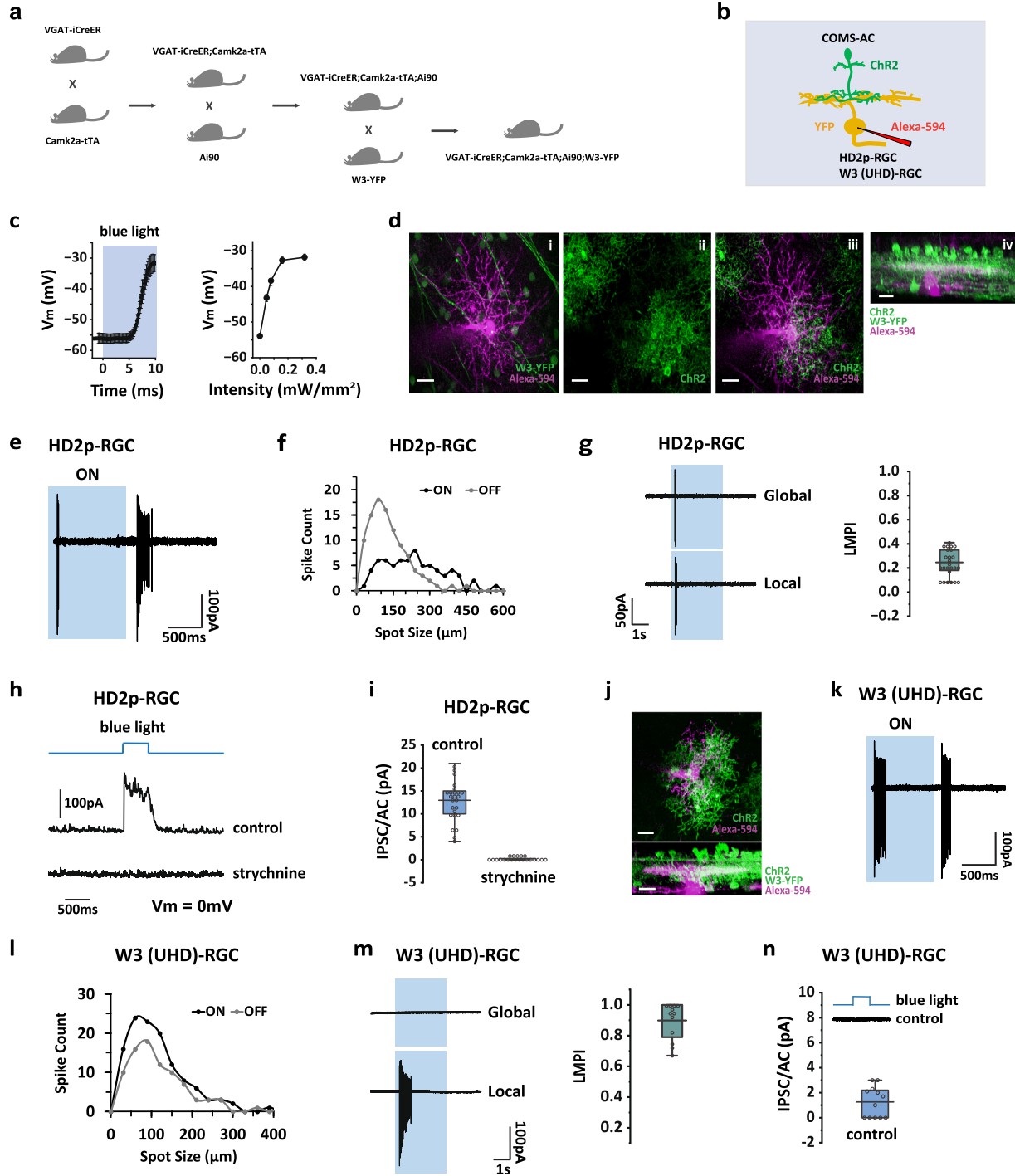

We then investigated the contribution of TH2-AC to the OMS in HD2p-RGC. By adding 10 μM C21 to the bath, we observed an approximately 6-fold increase in the spike rate of HD2p-RGC in response to global motion, while its response to local motion remained unaffected (Fig. 9e, f). Consequently, HD2p-RGC transitioned from an OMS insensitive RGC to an anti-OMS RGC with a preference for global motion over local motion, as evidenced by an LMPI of $-0.49 \pm 0.18$ ($n = 8$) (Fig. 9g). These results indicate that HD2p-RGC is inhibited by TH2-AC during global motion but not during local motion, suggesting a similar mechanism to that found in W3-RGC (UHD-RGC)[28].

VGlut3-AC uses both glutamate and glycine as neurotransmitters[14,15,23,46]. It provides excitatory input to W3-RGC

inactivation, we utilized the TH-Cre/TH-2A-FlpO intersection which removed labeling from SACs as well as many DACs, resulting in a more selective labeling of TH2-ACs (Supplementary Fig. 8). We then utilized RC::FPDi to express DREADD hM4Di to TH2-ACs (Fig. 9a) and labeled HD2-RGC by creating a quadruple mouse line: TH-Cre; TH-2A-FlpO; RC::FPDi; W3-YFP.

We examined the effect of TH2-AC inactivation using C21 (10 μM) on the receptive field of HD2p-RGC. Our findings showed that C21 significantly increased the OFF spiking rates of HD2p-RGC in spots larger than 60 μm diameter (Fig. 9b, c) and increased the ON spiking rates in all spot sizes tested (Fig. 9b, d). These results suggest that TH2-AC provides both ON and OFF inhibition to HD2p-RGC with different spatial properties.

**Fig. 5 | Optogenetic mapping of postsynaptic RGCs of COMS-ACs.** Quadruple transgenic breeding scheme (**a**) to label COMS-ACs with ChR2 and RGCs with YFP (**b**). **c** ChR2-expressing COMS-AC was targeted for whole-recording to determine the optogenetic stimulus intensities for -25 mV depolarization (Fig. 2h, i). Left: average trace of membrane potential responses to 0.16 mW/mm² stimuli, $n = 5$ cells. Right; summary of membrane potential responses to increased stimulus strengths, $n = 5$ cells. 0.16 mW/mm² stimuli which produced -23 mV depolarization were used for optogenetic mapping in the following experiments. The data are presented as mean ± SEM. **d** A post-synaptic HD2p-RGC recorded in a VGAT-iCreER;Camk2a-tTA;Ai90;W3-YFP retina. The recorded RGC was labeled with Alexa 594 and projected through the complete stack in (i) and (iii). (i) The recorded RGC overlapped with the somas of YFP-labeled RGCs in W3-YFP. (ii) COMS-ACs were labeled with ChR2-EGFP in VGAT-iCre;Camk2a-tTA;Ai90. (iii) The recorded RGC overlapped with the dendrites of COMS-ACs in the middle of the IPL. (iv) The side view of the retina shows the overlap between the dendrites of the recorded RGC and COMS-ACs. Scale bar, 20 µm. Experiments were replicated independently in at least 40 cells with similar results. **e** Responses of HD2p-RGC to a light spot (100 µm

diameter) on a dark background under cell-attached recording, showing OFF-dominant responses. **f** ON and OFF receptive fields of HD2p-RGC. **g** HD2p-RGC spiking during global and local motion, with LMPI shown on the right ($n = 25$ cells). **h** Optogenetic stimulation of COMS-ACs evoked an IPSC in the HD2p-RGC, which was blocked by strychnine. **i** Summary of evoked IPSCs in HD2p-RGCs, calculated as IPSC from a single COMS-AC, $n = 25$ cells. **j** A W3 (UHD)-RGC recorded in a VGAT-iCre;Camk2a-tTA;Ai90;W3-YFP retina. Experiments were replicated independently in at least 30 cells with similar results. Scale bar, 20 µm. **k** Responses of W3 (UHD)-RGC to a light spot (100 µm diameter) on a dark background, showing equivalent ON-OFF responses. **l** ON and OFF receptive fields of W3 (UHD)-RGC. **m** W3 (UHD)-RGC spiking during global and local motion, with LMPI shown on the right, $n = 12$ cells. **n** Optogenetic stimulation of COMS-ACs did not produce IPSCs in the W3 (UHD)-RGC, $n = 12$ cells. Top inset: an example IPSC recorded in W3 (UHD)-RGC during optogenetic stimulation. **g, i, m**, and **n** The box plots display the mean, 25th, and 75th percentiles, while the whiskers indicate the 1.5 interquartile range. Source data of (**c, g, i, m**, and **n**) are provided as a Source Data file.

(UHD-RGC) during local motion. Could it extend its influence on HD2p-RGC, providing either excitatory input (via glutamate release) or inhibitory input (via glycine release) during local motion? To examine the role of VGlut3-AC, we injected AAV2(YF4)-smCBA[47]-DIO-hM4Di into the eyes of VGlut3-Cre mice that were crossed with W3-YFP (Supplementary Fig. 9a). By using C21 (10 µM), we examined the effects of VGlut3-AC inactivation on EPSC and IPSC in HD2p-RGC. As shown in Supplementary Fig. 9b–m, C21 has no effect on EPSC and IPSC during either local motion (Supplementary Fig. 9b–g) or global motion (Supplementary Fig. 9h-m). These results suggest that VGlut3-AC is not actively involved in the OMS circuitry in HD2p-RGC. Our findings align with a recent 3DEM investigation which demonstrates the preference of VGlut3-AC to establish synaptic connections with W3 (UHD, 5ti) cells while avoiding interactions with HD2 (5so) cells (https://www.biorxiv.org/content/10.1101/2023.07.03.547571v1).

Based on our findings, we propose a model for the object motion sensitivity (OMS) circuit of HD2p-RGC, as shown in Fig. 9h. During object motion, HD2p-RGC is inhibited by two parallel AC pathways: one mediated by COMS-AC, which provides glycinergic inhibition during local motion, and the other mediated by TH2-AC, which provides GABAergic inhibition during global motion. The latter pathway is shared with W3 (UHD)-RGC. Therefore, the retina uses a shared TH2-AC circuit to control the responses of HD2p-RGC and W3 (UHD)-RGC to global motion but employs two opposing circuits to either inhibit (via COMS-AC) or enhance (via VGlut3-AC) their responses to local motion. The counterbalance between COMS-AC and TH2-AC results in the OMS insensitive characteristic of HD2p-RGC, while the cooperation between VGlut3-AC and TH2-AC leads to the OMS sensitive feature in W3 (UHD)-RGC.

## Discussion

OMS RGCs were first described in rabbit and salamander and include "ON Brisk Transient" cells and "ON−OFF Direction Selective" cells in the rabbit and "Fast OFF" cells in the salamander[26]. In the mouse, at least three RGC types (W3(UHD)-RGC, HD1-RGC and LED-RGC) show OMS[15,25,27,30]. On the other hand, the retina also contains RGC types that show much smaller or no difference between the global and local motion. While the presence or absence of OMS elements in RGC circuits may explain their OMS sensitive or OMS insensitive feature, our study reveals a more sophisticated mechanism that involves different combinations of OMS-driving and OMS-canceling interneurons. At least three parallel AC pathways (COMS-AC, VGlut3-AC, and TH2-AC) work in concert to regulate the OMS of RGCs. VGlut3-AC and TH2-AC promote OMS, while COMS-AC provides local motion inhibition and functions in opposition to VGlut3-AC and TH2-AC. COMS-AC acts as a counteractive domain that serves as a circuit breaker to cancel the OMS responses in selective RGC types, presenting a mechanism for

"redundancy-reduction". The discovery of the counteractive function of COMS-AC suggests that similar ACs may be employed in other feature detection pathways to further segregate feature selectivity among different RGC types and increase the neural coding complexity within the limited space of the IPL.

Modular organization is a defining characteristic of brain networks[48–50], as it enables specialized information processing, complex dynamics, and cost-efficient spatial embedding. The retinal IPL is divided into 10 strata, where -13 types of bipolar cells and 60 types of ACs form excitatory and inhibitory synapses with more than 40 types of RGCs, resulting in their unique response properties. However, it is not yet understood how different RGC types with similar dendritic stratifications acquire distinct responses from the shared strata. HD2-RGC and W3(UHD)-RGC are members of the "high-definition" (HD) family of RGCs[25]. They both have dendrites that stratify in the middle of the inner plexiform layer (IPL) and receive excitatory input from transient ON and OFF bipolar cells. They both respond to small spots and exhibit strong surround suppression, but they differ in various aspects, including the mechanisms of surround suppression, speed and tuning curve for responses to small moving objects, performance in tracking moving objects, and responses to object motion. These differences suggest that despite their similar dendritic stratifications, they have distinct wiring diagrams. HD2p-RGC closely resembles HD2-RGC, exhibiting strong similarities in terms of morphology, spiking patterns, and receptive fields (Fig. 5d-g). The LMPI calculated from local and global responses in HD2-RGC is marginally higher than in HD2p-RGC (0.45 vs. 0.29). However, it is important to note that this discrepancy is likely attributed to the distinct stimuli employed in these two studies. Our study demonstrates that HD2p-RGC and W3(UHD)-RGC use shared inhibition from TH2-AC during global motion and distinct inhibition/excitation during local motion to possess OMS insensitive or OMS sensitive feature, respectively. This mechanism is an example of "modular interneuron circuits." Similar circuits have been identified in looming detection, where W3(UHD)-RGC and OFF α RGCs combine shared excitatory input from VGlut3-AC with differing inhibition from TH2-AC and AII-AC to encode onset and speed, respectively[51]. It is becoming increasingly evident that modular combinations of ACs are commonly used to encode various visual features. The modular design of interneuron circuits in the retina could enhance coding complexity by effectively utilizing available synaptic connections to form flexible combinations of excitatory and inhibitory domains for specific tasks.

Are there other AC types in the retina that are also object motion sensitive? Our research has demonstrated that the OMS in COMS-AC does not originate from TH2-AC (Supplementary Fig. 2). Interestingly, VGlut3-AC also does not receive significant input from TH2-AC either[28]. Therefore, the mechanisms underlying the OMS of COMS-AC

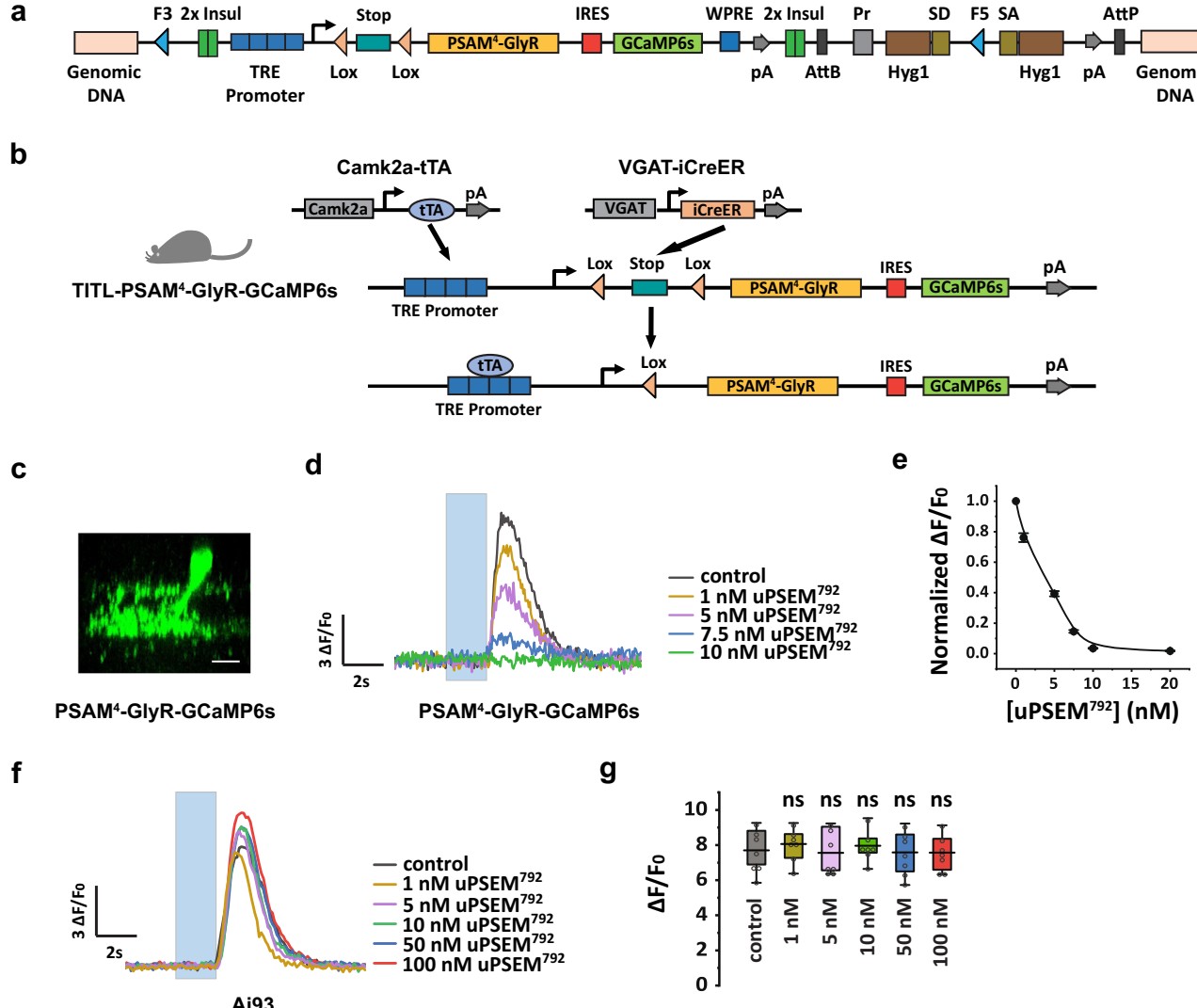

**Fig. 6 | TITL-PSAM⁴-GlyR-GCaMP6s mouse line enables Cre/tTA dependent chemogenetic silencing. a** Design of a Cre/tTA-dependent silencing mouse line, TITL-PSAM⁴-GlyR-GCaMP6s. **b** Breeding scheme for generating triple transgenic mice expressing PSAM⁴-GlyR-GCaMP6s in COMS-AC. **c** Expression of PSAM⁴-GlyR-GCaMP6s was observed in COMS-AC in the triple transgenic mouse line. Scale bar: 10 μm. Experiments were replicated independently in at least 40 cells with similar results. **d** GCaMP6s responses of COMS-AC to different concentrations of uPSEM⁷⁹², a PSAM⁴-GlyR agonist. **e** Dose-response curve of uPSEM⁷⁹² for PSAM⁴-GlyR-GCaMP6s in COMS-AC. N = 10 cells, error bars represent SEM. **f** GCaMP6f responses

of control COMS-AC (without PSAM⁴-GlyR) in VGAT-Cre;Camk2a-tTA;Ai93 mice to different concentrations of uPSEM⁷⁹². **g** uPSEM⁷⁹² had no effects on control COMS-AC without PSAM⁴-GlyR. N = 7 cells, ns: $p = 0.94$ for 1 nM, $p = 0.81$ for 5 nM, $p = 0.69$ for 10 nM, $p = 0.94$ for 50 nM, $p = 0.69$ for 100 nM, Wilcoxon Signed Rank test, two tailed. GCaMP6f and GCaMP6s responses were recorded in the distal dendrites with a spot of light (50 μm) at the center of the receptive field. The box plot displays the mean, 25th, and 75th percentiles, while the whiskers indicate the 1.5 interquartile range. Source data of (**e**) and (**g**) are provided as a Source Data file.

and VGlut3-AC require further investigation. It is possible that they share the same GABAergic inputs in global motion, and it would be intriguing to determine whether the GABAergic inputs originate from a type of polyaxonal ACs homologous to those in salamanders that are activated by movement in the surround[52]. TH2-AC is known to make synaptic connections with at least three additional RGC types, namely G4, G5, and G14[53], besides W3 (UHD-RGC)[54]. In this study, we found that HD2p-RGC (G14) is one of them based on its morphology. G4/G5 likely corresponds to tOFFα RGCs, as suggested by previous studies. Furthermore, SBEM analysis has revealed that COMS-AC (type H19[11]) forms synaptic connections with tOFFα RGCs (David Berson, personal communication). It has also been shown that tOFFα RGCs receive glutamatergic input from VGlut3-AC[51]. Investigating how the parallel pathways mediated by COMS-AC, VGlut3-AC, and TH2-AC contribute to the OMS in tOFFα RGCs would be of great interest.

While COMS-AC contributes to the center inhibition of HD2p-RGC (Fig. 7c) where the silencing of COMS-AC results in increased HD2p-RGC spiking during small spot stimulation. However, this inhibition is not sufficient to completely suppress HD2p-RGC spiking. One plausible explanation could be that a relatively low GlyT1 expression in COMS-AC compared with other glycinergic ACs[9,37], combined with its transient response to spot stimulation (Fig. 2h), results in a relatively weak and momentary inhibition in HD2p-RGC. This inhibition may not be strong enough to cancel the strong excitation experienced by HD2p-RGC during spot stimulation. The function of HD2-RGC is still unknown, although its performance is optimized for tracking small objects of 200 μm in diameter on the retina[25]. Additionally, the central projections of HD2-RGC remain a mystery. RGC axonal terminals labeled in W3-YFP mice are known to project to the superficial superior colliculus (sSC)[43], which likely include HD2-RGC axons. Trans-synaptic tracing with rabies virus has revealed that HD1-RGC and/or HD2-RGC

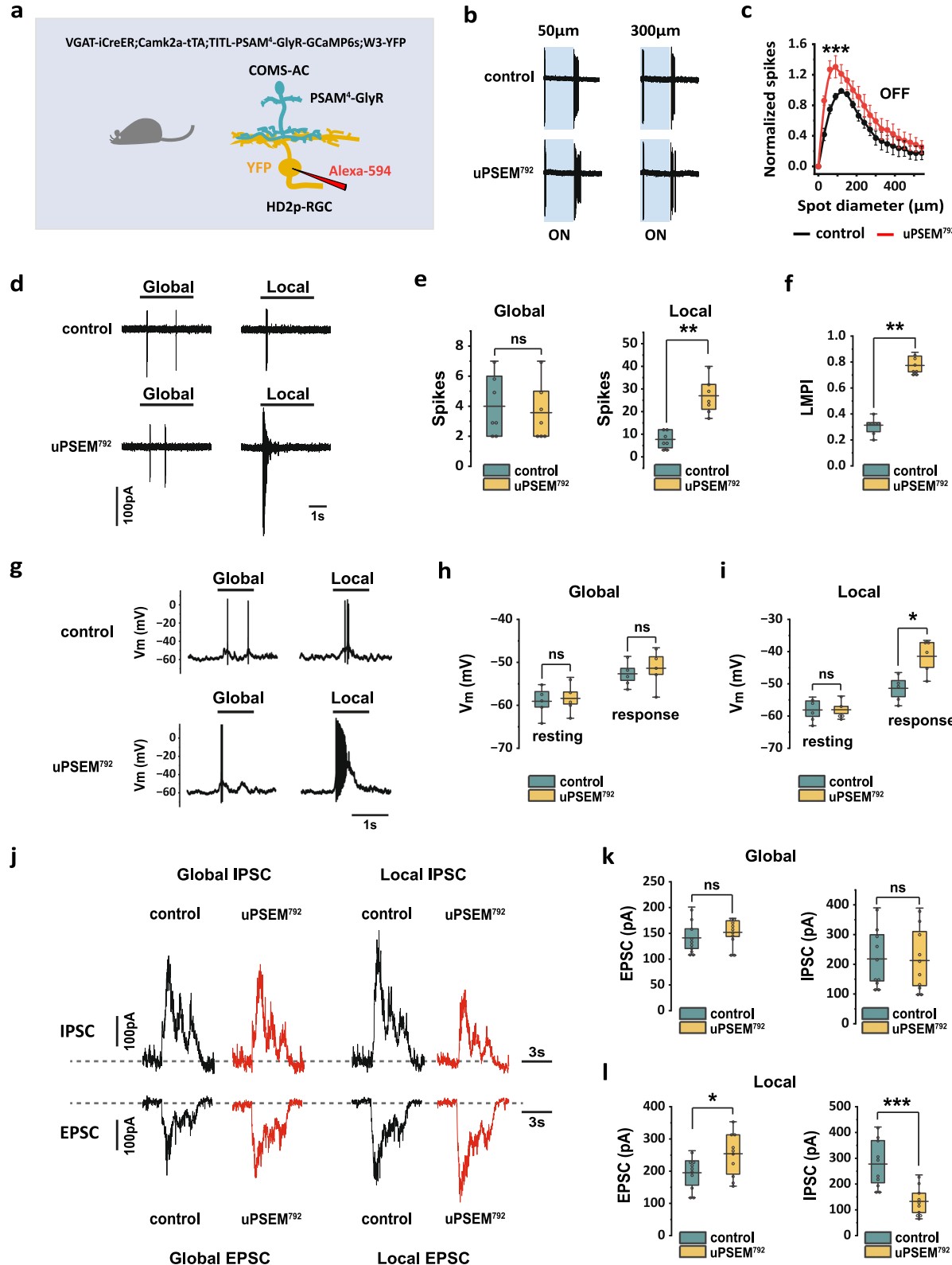

project to SC neurons innervating the pulvinar nucleus and the para-bigeminal nucleus[55]. Future experiments are needed to confirm the disynaptic HD2-RGC−sSC−pulvinar nucleus pathway and HD2-RGC−sSC−pulvinar nucleus−parabigeminal nucleus pathway and to deter-mine their behavioral significance.

In summary, this study uncovered an active circuit breaker, the COMS-AC, that controls object motion detection in the retina. Similar circuit breakers, together with "modular interneuron circuits" could be embedded into other retinal strata to expand their neural coding capacity.

## Methods

### Generation of TITL-PSAM[4]-GlyR-GCaMP6s mouse line

To create the TITL-PSAM[4]-GlyR-GCaMP6s mouse line, we obtained Ai99 ES cells containing docking sites for a replacement vector from the Allen Institute. We used the Ai62 plasmid (Addgene #61576) as a

**Fig. 7 | Silencing COMS-AC renders HD2p-RGC sensitive to object motion.**
**a** Schematic illustration for labeling COMS-AC with PSAM⁴-GlyR and HD2p-RGC with YFP. **b** uPSEM⁷⁹² increased the HD2p-RGC OFF spiking rate in response to spot stimulation at both 50 μm and 300 μm. **c** The OFF receptive field of HD2p-RGC in control and uPSEM⁷⁹² (10 nM). Responses in uPSEM⁷⁹² were normalized to the control. $N = 6$ cells, ***$p = 2.1E-4$, Wilcoxon Signed Rank test, two tailed. The data are presented as mean ± SEM. **d** HD2p-RGC spiking during global and local motion, with and without uPSEM⁷⁹². **e** uPSEM⁷⁹² significantly increased HD2p-RGC spiking rate during local stimulation, but not during global stimulation, $N = 7$ cells. ns: $p = 0.25$, Wilcoxon Signed Rank test, two tailed. **$p = 0.0078$, Wilcoxon Signed Rank test, one tailed. **f** uPSEM⁷⁹² increased the LMPI of HD2p-RGC. $N = 7$ cells. **$p = 0.0078$, Wilcoxon Signed Rank test, one tailed. **g** Membrane potential ($V_m$) of HD2p-RGC during global and local motion, before (control) and after the application of uPSEM⁷⁹². Global motion and local motion were applied for 1cycle (1 s/cycle).

**h** uPSEM⁷⁹² had no effect on resting potential (resting) and $V_m$ in response to global motion (response). $N = 5$ cells, ns: $p = 0.63$ for both resting and response, Wilcoxon Signed Rank test, two tailed. **i** uPSEM⁷⁹² produced depolarization in response to local motion (response), without changing resting potential (resting). $N = 5$ cells, ns: $p = 1$, Wilcoxon Signed Rank test, two tailed. *$p = 0.031$, Wilcoxon Signed Rank test, one tailed. **j** EPSC and IPSC were recorded in HD2p-RGC during global and local motion, with and without uPSEM⁷⁹². **k** uPSEM⁷⁹² had no effect on EPSC and IPSC in HD2p-RGC during global motion. $N = 10$ cells, ns: $p = 0.23$ for EPSC, $p = 0.77$ for IPSC, Wilcoxon Signed Rank test, two tailed. **l** uPSEM⁷⁹² increased EPSC and decreased IPSC in HD2p-RGC during local motion, $N = 10$ cells. *$p = 0.014$ for EPSC, ***$p = 0.00098$ for IPSC, Wilcoxon Signed Rank test, one tailed. **e, f, h, i, k,** and **l** The box plots display the mean, 25th, and 75th percentiles, while the whiskers indicate the 1.5 interquartile range. Source data of **c, e, f, h, i, k,** and **l** are provided as a Source Data file.

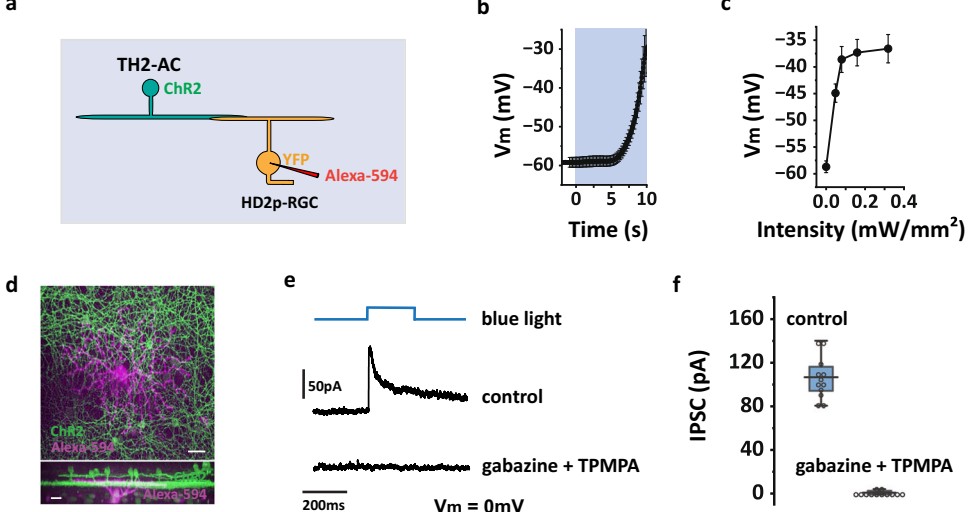

**Fig. 8 | TH2-AC provides GABAergic input to HD2p-RGC. a** Schematic illustration for expressing ChR2 in TH2-AC while recording from YFP labeled HD2p-RGC in TH-Cre; Ai32; W3-YFP mice. **b, c** ChR2-expressing TH2-AC was targeted for recording to determine the optogenetic stimulus strengths that produced a 15 mV–26 mV depolarization as measured from photoreceptor stimulation. **b** average trace of membrane potential responses to 0.16 mW/mm² stimuli, $n = 6$ cells. **c** summary of membrane potential responses to increased stimulus strengths, $n = 6$ cells. Optogenetic mapping in the following experiments used 0.16 mW/mm² stimuli which produced ~22 mV depolarization. **d** A recorded HD2p-RGC (magenta, labeled with Alexa-594) overlapped with ChR2-expressing TH2-ACs in a TH-Cre; Ai32; W3-YFP retina. Scale bar: 20 μm. Experiments were replicated independently in at least 20 cells with similar results. **e** Optogenetic stimulation of TH2-ACs evoked an IPSC in HD2p-RGC during whole-cell recording, which was blocked by gabazine and TPMPA. **f** Summary of ChR2-evoked IPSCs in HD2p-RGCs, $n = 12$ cells. The box plot displays the mean, 25th, and 75th percentiles, while the whiskers indicate the 1.5 interquartile range. **b** and **c** The data are presented as mean ± SEM. Source data of (**b, c,** and **f**) are provided as a Source Data file.

template and replaced the tdTomato sequence with PSAM⁴ GlyR-IRES-GCaMp6s using the PSAM⁴ GlyR plasmid obtained from Addgene (#119739). ES cell transfection, selection, and screening were performed following previously described[34]. After hygromycin selection, 12 ES cell colonies were chosen for genotyping, including five sets of primers designed to verify the 5′ and 3′ junctions and internal regions of the targeting constructs. Ten of the colonies were confirmed to have undergone the correct cassette exchange. Two colonies were selected for injection into C57BL/6 blastocysts to produce F0 generation mice, which almost 100% exhibited germline transmission and produced the F1 generation of TITL-PSAM⁴-GlyR-GCaMP6s mice.

## Mouse breeding and husbandry

All animal experiments were conducted in compliance with the Guide for the Care and Use of Laboratory Animals as adopted and promulgated by the US National Institutes of Health, and approved by the Institutional Animal Care and Use Committee of Northwestern University (protocol IS00011569). Adult mice of either sex, aged 8-12 weeks and maintained in a C57BL/6 congenic background, were used

for experiments. All animals were randomly assigned into the experimental and control groups, while maintaining a balance between females (~180 animals) and males (~180 animals). Animals had ad libitum access to food and water and were maintained on a regular 14-h light/10-h dark cycle. Ambient temperatures and humidity were $22 \pm 2$ °C and $50 \pm 10$%, respectively. The TH-2A-FlpO mouse line was kindly provided by Rajeshwar B. Awatramani at Northwestern University, and all other animal lines were obtained from The Jackson Laboratory: VGAT-iCreER (C57BL/6N-Tg(Slc32a1-icre/ERT2)3Gloss/J, JAX 016582), VGAT-Cre (Slc32a1^tm2(cre)Lowl/J, JAX 016962), CMV-Cre (B6.C-Tg(CMV-cre)1Cgn/J, JAX 006054), VGlut3-Cre (B6;129S-Slc17a8^tm1.1(cre)Hze/J, JAX 028534), Camk2a-tTA (B6.Cg-Tg(Camk2a-tTA)1Mmay/DboJ, JAX 007004), TH-Cre (B6.Cg-*7630403G23Rik*^Tg(Th-cre)1Tmd/J, JAX 008601), Ai93 (B6;129 S6-*Igs7*^tm93.1(tetO-GCaMP6f)Hze/J, JAX 024103), Ai90 (B6.Cg-*Igs7*^tm90.1(tetO-COP4*/EGFP)Hze/J, JAX 024100), Ai32 (B6.Cg-*Gt(ROSA)26Sor*^tm32(CAG-COP4*H134R/EYFP)Hze/J, JAX 024109), Ai65 (B6;129S-*Gt(ROSA)26Sor*^tm6S.1(CAG-tdTomato)Hze/J, JAX 021875), RC::FPDi (B6;129S6-*Gt(ROSA)26Sor*^tm9(CAG-mCherry,-CHRM4*)Dym/J, JAX 029040), W3-YFP (B6.Cg-Tg(Thy1-YFP)W3Jrs/J, JAX 033114). To activate iCreER, tamoxifen (75 mg/kg

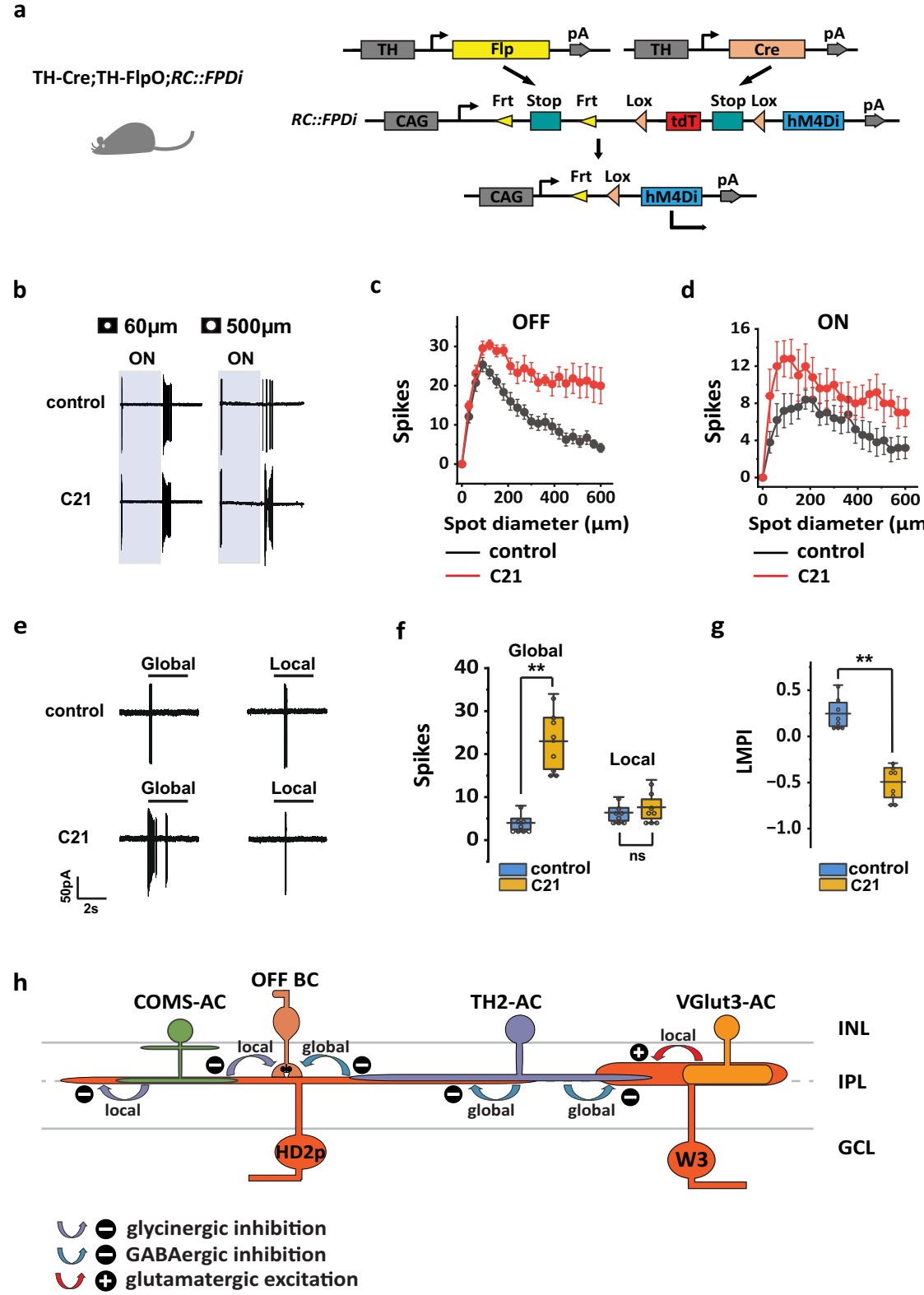

**Fig. 9 | Inactivation of TH2-ACs relieved surround inhibition and global inhibition in HD2p-RGC. a** TH-Cre; TH-2A-FlpO; RC::FPDi intersectional strategy for selective inactivation of TH2-ACs. **b** Compound 21 (C21), a hM4Di agonist, increased the HD2p-RGC OFF spiking rate in response to 500 μm spot stimulation, but not to 60 μm spot stimulation. Effects of C21 on OFF (**c**) and ON (**d**) receptive fields of HD2p-RGCs, n = 5 cells. **e** C21 increased the spiking rate of HD2p-RGCs in global motion, but not in local motion. **f** Effects of C21 on HD2p-RGC responses in global and local motion. N = 8 cells, ns: p = 0.23, Wilcoxon Signed Rank test, two tailed. **p = 0.0039, Wilcoxon Signed Rank test, one tailed. **g** C21 decreased the

LMPI of HD2p-RGC and rendered it global motion sensitivity. N = 8 cells, **p = 0.0039, Wilcoxon Signed Rank test, one tailed. **h** A comparison of the OMS circuitries in HD2p-RGC and W3 (UHD)-RGC. Both circuits use TH2-AC to inhibit responses to global motion, but the HD2p-RGC circuit uses COMS-AC to inhibit while the W3 (UHD)-RGC circuit uses VGlut3-AC to enhance responses to local motion. Bipolar cell inputs to W3 (UHD)-RGC are not included in the diagram. **f** and **g** The box plots display the mean, 25th, and 75th percentiles, while the whiskers indicate the 1.5 interquartile range. **c**, **d** The data are presented as mean ± SEM. Source data of (**f**) and (**g**) are provided as a Source Data file.

body weight) was injected intraperitoneally after postnatal 100 days for 3 weeks. Seven days after the last induction, mice were subjected to recording.

## Viral packaging and injection

AAVs were generated and purified following established protocols[56]. To summarize, the AAVs were synthesized through Polyethylenimine (PEI) transfection of HEK293T/17 cell (ATCC, CRL-11268) cultured in adherence. This process incorporated AAV cis, AAV trans, and adenovirus helper plasmid pAdΔF6. After 72 h of transfection, the AAVs were collected from both cell pellets and media, then subjected to purification through iodixanol gradient ultracentrifugation. Subsequently, the viruses were concentrated and formulated using PBS.

AAV injections were performed on 7- to 8-week-old mice. 1 μl of *AAV2 (YF4)- smCBA-DIO-hM4Di* ($1 \times 10^{13}$ genome copies/ml) was injected into the eyes of VGlut3-Cre;W3-YFP mice. Recordings were performed 5 weeks after injection.

## Immunohistochemistry and imaging

For immunohistochemistry, adult mice were euthanized, and their eyes were fixed with 4% paraformaldehyde. The retinas were dissected from the eyecup and washed with a modified phosphate buffer (PB) containing 0.5% Triton X-100 and 0.1% NaN3 (pH 7.4) six times for 30 min each. The retinas were then blocked for 2 days in modified PB containing 5% donkey serum and incubated with primary antibodies for 5 days at 4 °C. After washing, the retinas were incubated with donkey secondary antibodies for another 2 days at 4 °C. Finally, the retinas were washed with PB and mounted with Vectashield (Vector Laboratories) on coverslips. The primary antibodies used were as follows: chicken anti-GFP (1:1000, Abcam ab13970), rabbit anti-RFP (1:500, Rockland, 600-401-379), goat anti-acetyltransferase (1:500, Millipore AB144P), rabbit anti-GlyT1 (1:200, antibodies-online, ABIN1841935), rabbit anti-PPP1R17 (1:1000, Millipore Sigma HPA047819), rabbit anti-Tyrosine hydroxylase (1:500, Novus Biologicals, NB300-110). Secondary antibodies were conjugated to Alexa Fluor 488, Cy3, or Cy5 (Jackson ImmunoResearch). All secondary antibodies were used at a dilution of 1:200.

## Confocal imaging

Confocal imaging was performed using a Zeiss LSM-510 Meta confocal microscope with 25× or 63× objectives, and the images were processed using LSM Image software, Image J, and Photoshop. Z-stack images were obtained with a 63× objective at 0.25 mm intervals. To measure the size of the dendritic field, we drew a convex polygon connecting the outermost tips of the dendrites, and the area within this contour was measured. The diameter of the dendritic field was calculated from the measured area by assuming the dendritic field is circular. Soma diameter was calculated in the same way. We determined the stratification levels from the upper and lower boundaries of GFP-labeled arbors relative to the choline acetyltransferase (ChAT)-positive bands (60% and 27% of the IPL) in the XZ plane. The data are presented as mean ± standard deviation (SD). The DRPs were computed following the definition outlined in a previous study[57], utilizing the X−Y coordinates of soma locations.

## Visual stimulation

Visual stimuli were generated using MATLAB (MathWorks) and projected onto the photoreceptors using a digital projector system (TI LightCrafter4500 with modified LEDs) via a substage condenser. To measure spatial tuning, spot stimuli with varying diameters were presented with 1 Hz temporal square-wave modulations (100% Michelson contrast) centered on the receptive field, with an average intensity of ~1500 rhodopsin isomerization/cone/s. To measure object motion sensitivity, square wave gratings (bar width ~45 μm) were presented in the center (75–120 μm diameter) and surround, either moving together (global motion) or with only the center grating moving (local motion) at a speed of 100 μm/s. Each cycle of grating motion lasted 1 s, and a total of 3 cycles were applied for global or local motion. A gray annulus (30 μm in width) was placed outside the center to separate movement in the center and surround.

## Two-photon GCaMP6 imaging

Mice were dark-adapted for at least 2 h. Eyes were enucleated and the retinas were dissected under infrared illumination. The retinas were mounted vitreal side up in the recording chamber and continuously perfused with oxygenated Ames medium at room temperature. Imaging was conducted using a Thorlabs multiphoton microscope equipped with a Mai Tai DeepSee ultrafast laser tuned to 940 nm, and label cells were visualized using a 20× objective (XLUMPLFLN, 1.0 numerical aperture, Olympus). To separate GCaMP6f signal acquisition from light stimulation, the projector LEDs (460 nm) were electronically gated by a copy of the resonant scanner trigger signal at 8 kHZ, such that the GCaMP6f signal was acquired during a forward scan sweep while the image was projected onto the retina during the return (discarded or unmonitored) sweep. We probed the center of the cell's receptive field using a 30 μm light spot generated by the projector, which was then moved to various positions on the retina. The central location was subsequently saved for future recordings. ThorImageLS software (Thorlabs, Inc.) was used for Imaging acquisition. Fluorescence values were obtained by averaging across the dendritic fields and analyzed using HCImage (Hamamatsu Photonics). Pharmacological reagents, including UBP310 (25 μM; Tocris Bioscience), strychnine (5 μM; Sigma Millipore), gabazine (50 μM; Sigma Millipore), MFA (100 μM; Sigma Millipore), TPMPA (50 μM; Tocris Bioscience), and uPSEM792 (1–100 nM, Tocris Bioscience), were added to the bath solution.

We calculated the LMPI in accordance with a prior study[28].

$$LMPI = (R_{local} - R_{global})/(R_{,local} + R_{,global})$$

$R_{local}$: responses to local stimulation, $R_{global}$: responses to global stimulation

## Electrophysiology and optogenetics

For two-photon imaging, an external solution of oxygenated Ames medium was used. GCaMP6f-labeled COMS-ACs or YFP-labeled RGCs in W3-YFP mice were targeted under two-photon illumination (940 nm). Whole-cell recordings were performed using patch pipettes with tip resistance of 5–7 MΩ for RGCs and 10–12 MΩ for ACs. The membrane current or potential was amplified, digitized at 10–20 kHz using an Axopatch 700B amplifier and Digidata 1440 A Digitizer, and stored for analysis using pClamp 10.0 (Molecular Devices). For current-clamp recording, intracellular solution containing (in mM): 125 K-gluconate, 10 NaCl, 1 MgCl₂, 10 EGTA, 5 HEPES, 5-ATP-Na, 0.1 GTP-Na (280 mOsm; pH adjusted to 7.4 with KOH) was used. The intracellular solution for voltage-clamp recording was composed of (in mM): 105 Cesium methanesulfonate, 10 TEA-Cl, 20 HEPES, 10 EGTA, 2 QX-314, 5 Mg-ATP/ 0.5 Tris-GTP (280 mOsm; pH adjusted to 7.4 with CsOH). To visualize RGC dendrites, 0.02 mM Alexa 594 was included in the intracellular solution. To enhance RGC labeling for post-recording analysis, 3% w/v Neurobiotin was added to the pipette solution in a subset of recordings. Absolute voltage values were corrected for a liquid junction potential of −12.8 mV. EPSCs and IPSCs were isolated by holding the membrane potential at −70 mV and 0 mV, respectively.

For activation of ChR2-expressing COMS-AC or TH2-ACs, a flash of blue light (455–495 nm, $1.6 \times 10^{-4}$ W/mm²) was used through a 20× Olympus XLUMPLFLN objective (1.00 NA). To block glutamatergic transmission from photoreceptors to bipolar cells, a mixture of blockers containing L-AP4 (20 μM; Tocris Bioscience), CNQX (25 μM;

Tocris Bioscience), UBP310 (20 μM; Tocris Bioscience), and D-APV (50 μM; Tocris Bioscience) was used during recordings.

## Quantification and statistical analysis

Experiments were conducted using mice of both sexes. OriginPro 2021(OriginLab Corporation) was used for statistical analysis. The data are presented as mean ± SEM unless stated otherwise. The box plot displays the mean, 25th, and 75th percentiles, while the whiskers indicate the 1.5 interquartile range. The Wilcoxon Signed Rank test was used to compare related groups, and statistical significance was accepted at $p < 0.05$.

## Reporting summary

Further information on research design is available in the Nature Portfolio Reporting Summary linked to this article.

## Data availability

Source data are provided with this paper. The data generated in this study are available in the paper, supplementary information, and source data. Source data are provided with this paper.

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

## Acknowledgements
We thank Dr. Rajeshwar B. Awatramani (Northwestern University) for providing TH-2A-FlpO mouse line and the Allen Institute for providing the Ai99 ES cells. We thank Dr. Maureen McCall (University of Louisville) for sending W3-YFP mouse line. This work was supported by NIH grants R01 EY030169 (Y.Z.), R01 EY018204 (S.H.D. and Y.Z.), R01 EY032506 (Y.Z. and S.H.D.), R01 EY012141 (S.H.D.), F31 EY031985 (A.J.), Whitehall Foundation Grant (Y.Z., 2017-05-20), and Research to Prevent Blindness.

## Author contributions
Y.Z. designed the experiments. Y.Z. and S.H.D. supervised the project. A.J., S.D., and S.C. performed GCaMP6f imaging and patch-clamp recording. J.X. generated TITL-PSAM4-GlyR-GCaMP6s mouse line. J.X. and Y.Z. provided transgenic colony management and genotyping. Y.Z., A.J., and D.F. conducted data analyses. Y.Z. and S.H.D. wrote the manuscript with contributions from all coauthors.

## Competing interests
The authors declare no competing interests.
