## [Peer Review File · Nature Communications]

Modular interneuron circuits control motion sensitivity in the mouse retinaEditorial Note: Parts of this Peer Review File have been redacted as indicated to maintain the confidentiality of unpublished analyses and related comments, and to remove third-party material.

REVIEWER COMMENTS

Reviewer #1 (Remarks to the Author):

The manuscript "Modular interneuron circuits control motion sensitivity in the mouse retina" by Jo et al. explores the role of specific intermingled interneuron types in the retina in separating out different types of visual information for transmission to the brain. Using advanced intersectional genetic strategies, optogenetics, and pharmacogenetics, the authors explore the function and circuit role of an amacrine cell type, CK2-AC1, that synapses with ganglion cells in a retinal layer associated with object motion sensitivity. Their work shows how this amacrine cell type is object motion sensitive and provides inhibition to a downstream cell type, the HD2 cell, setting up the HD2 cells to be "high definition" and not respond strongly to object motion.

While this study uses exciting methods to explore the functional role of amacrine cells in retinal circuits, the motivation and interpretation of some of the experiments are problematic. In addition, further quantification of the intersectional strategy is necessary to be certain that the optogenetic and pharmacogenetic manipulations are restricted to a single cell type.

Major comments

1. It is unclear from the data shown in Fig. 1 if the strategy of injecting tamoxifen after P100 actually restricts the mouse line to labeling a single cell type. This is important for being able to interpret the optogenetic and pharmacogenetic manipulations later in the paper. The normalized density in Fig 1Eiv shows that there is less overlap between cells using this strategy, but this could also result from sparser labeling of a heterogeneous population of multiple cell types.

Quantification of where the cells in these mice stratify in the inner plexiform layer would be better evidence that no CK2-AC2 cells remain, as well as a population-level quantification of the PPP1R17 immunohistochemistry results. The authors should include these analyses.

2. I found the motivations for this study confusing. First, the authors set up their introduction to be about specialization in stratification in the retina and in particular the strata where the W3 cells are located along with two HD cell types. Then in the results the focus on an intersectional strategy for labeling amacrine cells with no clear goal in mind. I assume that the authors set out to identify amacrine cell types co-stratifying with W3 and HD cells, but they should make this clear at the start, and generally make the study's goals clear throughout.

3. The interpretation of the pharmacology results, in particular the effect of TPMPA, is problematic. Typically, GABA_A receptors are associated with bipolar cells because bipolar cells widely express them on both dendrites and axons. Indeed, some recent work has shown that bipolar cell motion properties may depend on GABA_A-mediated inhibition (Strauss et al Nature Communications 2022). According to transcriptomic data (Yan et al J Neurosci 2020), GABA_A receptors are expressed in only one amacrine cell type, cluster 56, which is a GAD⁺ type, not the type studied here. Thus, the interpretation of the voltage clamp recordings in Fig 2, that there is a direct GABA_A-mediated input to the CK2-AC1 cell type, is unlikely. For instance, the result that TPMPA leads to increased responses to global motion in CK2-AC1 may simply result from changes in the bipolar cell center-surround receptive fields. The authors should demonstrate that CK2-AC1 cells express GABA_A receptors or consider revising their interpretation of their results to an entirely presynaptic mechanism, with effects primarily on bipolar cells.

4. The characterization of the HD2 cell as a non-object motion sensitive cell seems somewhat inaccurate. In the paper that characterized this cell type (Jacoby and Schwartz, J Neurosci 2017), this cell type responds better to differential motion than to global motion (though not as strongly to either than to a "center only" visual stimulation). The authors should clarify in the discussion how their stimuli and results relate to the original characterization of this cell type and consider changing their description of the cell type to be a more accurate description of this cell type's preferences.

Minor comments

1. In the first paragraph of the introduction, the authors discuss the "current view" that retinal strata are specialized for encoding different stimulus properties. I was not aware that this was a

generally held view. The authors should cite their source for this.

2.The authors do not mention the role of upstream bipolar cells in their introduction or discussion of mechanisms for object motion sensitivity, even though bipolar cells may play an outsized role in setting up OMS in the retina (see Gaynes et al, Nature Communications 2022).

3.At the end of the first paragraph of the results, the authors mention several amacrine cell types from different classification systems that may correspond to the types identified with the intersectional strategy. Several acronyms are not defined in these sentences and the different classifications are not well-identified (one is EM and the other is transcriptomic). Displaying the EM data next to the cells identified using the intersectional strategy would be helpful to see in a figure.

4.The authors should include the GCaMP isoform in all mentions of this in the manuscript – whether it is '6f' or '6s'.

5.The figures are confusingly laid out, with panels for similar measurements in random spatial locations. For instance, in Figure 4, the same recordings and summary data are made in HD2 and W3 cells but the panels are not in a column as they could be to facility comparison. In addition, several of the electrophysiology traces did not print well and seem to require further down-sampling for display.

Reviewer #2 (Remarks to the Author):

The mammalian retina, like other brain regions, comprises >100 types of neurons. Highly specialized connectivity within and between types plays an important role in diversifying visual signals before leaving the optic nerve. This article by Jo and colleagues utilizes a powerful intersectional genetic approach for exploring connectivity between retinal ganglion cells and retinal interneurons. They identify 2 interneurons for which very little is known: CK2-AC1 and CK2-AC2, and are able to label one or both with various reporters and ChR2. Receptive field properties were pursued with selective GCamp6 expression in CK2-AC1. Spots of light revealed a preference for light decrements (OFF), and drifting grating revealed a strong preference for local movement over global movement. Voltage clamp recordings showed that EPSCs were reduced by global motion whereas IPSCs were enhanced. TPMPA, a GABA_AR antagonist, eliminated this effect. The authors then compare activity of two co-stratifying RGCs in response to local and global motion: the UHD and HD2. The UHD has a strong preference for local motion whereas the HD2 responds similarly to both local and global motion. ChR2 expression in CK2-AC1 revealed direct glycinergic connections between cells, and PSAM expression and exogenous modulation revealed a stronger response to local motion. Furthermore, the authors use their intersectional strategy to identify and modulate a wide-field amacrine cell type that inhibits the HD2, via GABA_ARs, during global motion. This is an impressive set of results, both technically and scientifically, and they should appeal to both retinal specialists and neuroscientists in general.

I see only two deficits in the study: 1) lack of identification of the WF AC that inhibits CK2-AC1, and 2) lack of RF field mapping for decrement steps, which the cells strongly prefer. Amacrine cell connectivity is notoriously difficult to identify and study, so the progress made here is, in my opinion, sufficient for publication even in the absence of WF AC identification. Im not really sure why the cells were not probed with decrement steps (from a mean), thus the Off responses shown here are really rebound responses.....this should be clarified explicitly in the text.

Minor:

1) I find the CK2-AC1(COMS-AC) notation to be a burdensome and a little confusing. Why use two names here? I suggest defining something upfront, and simplifying throughout the ms.

2) In a few places the authors say that the CK2-AC1 encodes OMS, but I would argue this is incorrect. The CK2-AC1 is object motion sensitive, and encodes information about local object motion.....it doesn't really encode OMS.

3) p.10 Redundant sentence.....To examine whether TH2-AC makes synaptic connections with HD2-RGC, we expressed ChR2 in TH2-AC while labeling HD2-RGC with YFP in TH-Cre;Ai32;W3-YFP mice. To investigate the presence of synaptic connections between TH2-AC and HD2-RGC, we

utilized optogenetics by expressing ChR2 in TH2-AC while labeling HD2-RGC with YFP in THCre; Ai32;W3-YFP mice (Fig. 7A).

4) p.10 '0.16mV/mm²'I believe you meant to use mW instead of mV, please double check

5) p.12 OMS gets defined again, but is now referring to the encoding of non-OMS...this is confusing. I would argue this should be described as Object Motion Insensitive.

6) Im a little surprised that the HD2-RGC has such a strong OFF response to a spot of light when CK2-AC1 has such a strong preference for OFF stimuli too. This seems like a feature that might also have cancelled, but it doesn't. I suggest adding a couple sentences to the discussion regarding this point.

7) Im not sure why all panels in Figure 4D are labelled with HD2-RGC. Seems unnecessary, and the image in Figure 4Dii , for example, doesn't have any labelled HD2s.

Small note: This reviewer really appreciates the control experiment in Figure 5. Sometimes exogenous compounds have unintended effects, and this was a clever way to control for this issue.

Reviewer #3 (Remarks to the Author):

Summary

This goal of this paper is to understand how signals from functionally distinct interneurons are mixed on physically proximate synaptic targets to generate different responses. The authors use the retina as a model and take aim at a pair of ganglion cells whose dendrites overlap extensively with amacrine cells (VG3 and CAII) whose signals are vital for object motion selectivity. One ganglion cell, called W3 sums input from VG3 and CAII to become object motion selective –it responds when motion in its center leads or lags motion in its surround. The other cell, called HD2, is the focus of this manuscript –its dendrites are entangled with VG3, CAII, and W3, but it does not show object motion selectivity. Using a series of elegant genetic experiments and electrophysiological analyses, the authors learn that HD2 receives inhibition from a newly identified AC type, CK2-AC1, which silences this RGC's object motion responses.

Explaining RGC visual responses in terms of their amacrine and bipolar inputs is a major, and as yet, incomplete goal. We know a great deal about direction selective retinal circuits, but know very little about any other retinal circuits. The results in this study reveal connectivity in a new circuit and illustrate a novel finding –that distinct amacrine-RGC connectivity patterns within a retinal neuropil layer create distinct visual responses. Specifically, the authors show a pair of circuits occupying the same retinal neuropil layer differ in their preference for local motion because one of these circuits (HD2) receives input from a novel amacrine type (CK2). There are also important tools devised here –the access to CK2-ACs is likely to be important to the retina field while the TITL-PSAM line is likely to be useful to neuroscientists in general. I thought the experiment using PSEM/PSAM to 'unmask' local motion responses on HD2 RGCs was particularly revealing.

I liked this study and the authors are to be commended on a tough but revealing series of experiments. However, there were many typographical errors, nomenclature errors, and missing pieces of data which should be addressed.

Major.

1) W3-RGCs are concentrated in the ventral half of the mouse retina (bleckert et al., Zhang et al.), with a possible interpretation being that they were specialized to sense aerial predators. Given the finding that HD2-RGCs are non-OMS due to CK2-AC1, it would be interesting to know the distribution of these cells are across the retina. It is probably too hard for HD2, but given the stains for CK2-ACs in Fig.1, I wonder if the authors could report the absolute density of these two types and their regional bias (ie: more in dorsal retina or ventral retina, etc). I imagine they have this data from the VGAT-Cre/Camk2a-tta/Ai93 cross with a ppp1r17 stain.

2) The comparison between global and local motion is used to measure the contribution of certain

kinds of input (ie: gabazine) or between RGCs (W3 vs HD2). The stimulus listed as 'local' would really be a differential motion stimulus according to the original definition of OMS (Olveczky et al., 2003). Local motion would be a stimulus presented to the receptive field center only. The follow-up paper describing HD2 (Jacoby et al. 2019) used a variant of this stimulus which moved a central texture in a different direction than the surround texture (fig9 of Jacoby et al.). Given that the authors believe that CK2 provides input to HD2, it would be nice to see its responses to this texture stimulus to register the results seen here with previous work.

3) Are the responses of the two dendritic strata of CK2-AC1 the same for local, global, and differential stimuli? I assume both data were grouped for subsequent experiments, but it would have been nice to see a bit more analysis of this dataset that justifies that recordings from different strata can be combined. If this was not done, then how was imaging depth standardized?

4) It is cool that CK2-AC1 blunts the local motion response on HD2, but as the authors mention in the discussion, the 'feature' that HD2 responds best to is unknown. As discussed, it seems as though the HD2-RGC does not respond well to any stimuli. I think this discussion paragraph can be improved.

5) The data in Fig6G and J is interesting and show that inhibiting the CK2-AC1 causes HD2-RGCs to become sensitive to local motion. As written in the introduction, VG3-ACs give rise to the local (and object) motion response of W3 – does the presence of a local motion response in HD2 mean that it also receives input from VG3? From the diagram in Fig.8, it seems as though the authors do not believe this is the case. Why? A sentence or two on this point would help a reader synthesize these results with the previous work on object motion.

6) The authors make a strong case that CK2-AC1 inhibition to HD2 blunts their local motion response. But given that VG3 provides glycinergic inhibition to suppressed-by-contrast RGCs, could it also inhibit HD2-RGCs ? A sentence or two about this in the discussion would help.

7) Some quantification of the synaptic latency in the optogenetic studies would have been nice to see.

8) There are many typographical errors in the text and figures. These need to be corrected.

9) The methods are missing details about how LMPI was computed, how DRPs were computed, and other forms of analyses. These should be added. There's also little information about how fast the gratings were moving for local and global stimuli. Some sentences should also be written about how the receptive field centers of the CK2-ACs were found.

10) All throughout, I think the authors have mistaken W3 for UHD when it should LED – Jacoby et al., show that UHD and HD2 are not object motion sensitive. Related to point 3 above, it is the differential motion response (together with size/speed tuning that distinguish these 4 cells).

Minor

1) "These two types occurred at a ratio ranging from 4:1 to 9:1." What is the definition for this ratio, I assume CK2:CK1 but I am not sure. Can you clarify.

2) reference is missing : "Additionally, CK2-AC1 showed a strong resemblance to nGnG1 and expressed PPP1R17 immunostaining (Fig. 1K), which is also expressed in nGnG1 and nGnG3." I suspect this should be PMID: 32457074.

3) "...while MAC 22 corresponds to nGnG3." What is MAC? By its position in the sentence I'd guess it is a typo that should read CK2-AC2?

4) Fig.1E. There are two "ii' panels

5) what does this mean: "we investigated how CK2-AC1 responds to visual stimuli in different layers and what are its receptive field properties at each level." I think you mean "we investigated

how visual responses were organized across the laminar depth of CK2-AC1 dendrites.

6) "These results suggest that the majority of inhibition to CK2-AC1 originates from GABAergic wide-field ACs and is mediated by GABAC receptors." There are several kinds of medium field ACs, how do the authors know it is wide? Some clarification needed here.

7) how is LMPI calculated? I could not find a section of the methods with this information

8) "TH2-AC co-stratifies with CK2-AC1 and demonstrates varying kinetics in response to global and local motions 17." Motions should be "motion"

9) there are several references missing from this sentence: "The retinal inner plexiform layer (IPL) is divided into 10 strata, where ~13 types of bipolar cells and 60 types of amacrine cells (ACs) form excitatory and inhibitory synapses with more than 40 types of retinal ganglion cells (RGCs), resulting in their unique response properties."

10) "These results suggest that the majority of inhibition to CK2-AC1 originates from GABAergic wide-field ACs and is mediated by GABAC receptors." There are several kinds of medium field ACs, how do the authors know it is wide? Some clarification needed here.

11) Figure 4G - "Glocal" should read "Global" I think.

12) "These findings suggest that GABAergic wide-field ACs provide both presynaptic and postsynaptic GABAC-mediated inhibitions to CK2-AC1 (Fig. 2N)." Inhibition is pluralized here and should be singular.

13) "These results suggest that during global motion, CK2-AC1 receives strong presynaptic and postsynaptic inhibitions that together produce an overall hyperpolarization." Inhibition is pluralized here and should be singular.

RESPONSE TO REVIEWERS

We would like to express our gratitude to the reviewers for their thoughtful comments and constructive suggestions for improvement. We greatly appreciate the overall enthusiasm for the manuscript expressed by the reviewers. They have identified important areas that require further refinement. In response to their critiques, we have made significant revision to the manuscript, which encompasses both additional experiments and written content. We believe that the reviewers' critical instructions have significantly improved the quality of our manuscript. In the following section, we provided detailed point-by-point responses to the reviewers' comments. Meanwhile, In the revised manuscript all the changed in text are highlighted.

REVIEWER COMMENTS

Reviewer #1 (Remarks to the Author):

The manuscript "Modular interneuron circuits control motion sensitivity in the mouse retina" by Jo et al. explores the role of specific intermingled interneuron types in the retina in separating out different types of visual information for transmission to the brain. Using advanced intersectional genetic strategies, optogenetics, and pharmacogenetics, the authors explore the function and circuit role of an amacrine cell type, CK2-AC1, that synapses with ganglion cells in a retinal layer associated with object motion sensitivity. Their work shows how this amacrine cell type is object motion sensitive and provides inhibition to a downstream cell type, the HD2 cell, setting up the HD2 cells to be "high definition" and not respond strongly to object motion.

While this study uses exciting methods to explore the functional role of amacrine cells in retinal circuits, the motivation and interpretation of some of the experiments are problematic. In addition, further quantification of the intersectional strategy is necessary to be certain that the optogenetic and pharmacogenetic manipulations are restricted to a single cell type.

Major comments

1. It is unclear from the data shown in Fig. 1 if the strategy of injecting tamoxifen after P100 actually restricts the mouse line to labeling a single cell type. This is important for being able to interpret the optogenetic and pharmacogenetic manipulations later in the paper. The normalized density in Fig 1Eiv shows that there is less overlap between cells using this strategy, but this could also result from sparser labeling of a heterogeneous population of multiple cell types. Quantification of where the cells in these mice stratify in the inner plexiform layer would be better evidence that no CK2-AC2 cells remain, as well as a population-level quantification of the PPP1R17 immunohistochemistry results. The authors should include these analyses.

We thank the reviewer for the important suggestion. In response, we have included Supplementary Figure 1b to provide evidence that the labeled cells exclusively stratified within the two bands where CK2-AC1s (now named COMS-ACs) stratify, while avoiding the stratification levels of CK2-AC2. Additionally, we have demonstrated in Supplementary Figure 1c and 1d that all the labeled cells exhibit positive PPP1R17 immunostaining, which further confirms that COMS-AC is the sole cell type labeled using this approach.

2.I found the motivations for this study confusing. First, the authors set up their introduction to be about specialization in stratification in the retina and in particular the strata where the W3 cells are located along with two HD cell types. Then in the results the focus on an intersectional strategy for labeling amacrine cells with no clear goal in mind. I assume that the authors set out to identify amacrine cell types co-stratifying with W3 and HD cells, but they should make this clear at the start, and generally make the study's goals clear throughout.

We apologize for any confusion regarding the motivation and rationale behind our experiments. This project is 'bottom-up' and primarily motivated by the discovery of a new AC type. Given our limited knowledge of AC types, we initiated our research by searching for new types and subsequently conducting investigations into their functions. This approach is probably more efficient than specifically seeking an AC type that stratifies within a particular stratum or serves a predefined function. COMS is the first AC type for which we successfully achieved single-cell labeling. Interestingly, it co-stratifies with HD2-RGC and W3-RGC and exhibits a noteworthy OMS feature. Upon making these observations, we focused on its interactions with HD2-RGC and W3-RGC.

When preparing the manuscript, we started with a general overview of retinal cell types and the challenge to study amacrine cell types in the first paragraph. Subsequently, in the second paragraph, we introduced the OMS circuits to provide readers with a foundational understanding. In the third paragraph, we integrated our experimental approach with our results.

In the revision, we have changed the first paragraph and shift it more towards the acknowledgment of our limited knowledge regarding the functions of amacrine cells. With these revisions, we believe the manuscript is now more clearly structured and conveys our intentions more effectively.

3.The interpretation of the pharmacology results, in particular the effect of TPMPA, is problematic. Typically, GABA_A receptors are associated with bipolar cells because bipolar cells widely express them on both dendrites and axons. Indeed, some recent work has shown that bipolar cell motion properties may depend on GABA_A-mediated inhibition (Strauss et al Nature Communications 2022). According to transcriptomic data (Yan et al J Neurosci 2020), GABA_A receptors are expressed in only one amacrine cell type, cluster 56, which is a GAD⁺ type, not the type studied here. Thus, the interpretation of the voltage clamp recordings in Fig 2, that there is a direct GABA_A-mediated input to the CK2-AC1 cell type, is unlikely. For instance, the result that TPMPA leads to increased responses to global motion in CK2-AC1 may simply result from changes in the bipolar cell center-surround receptive fields. The authors should demonstrate that CK2-AC1 cells express GABA_A receptors or consider revising their interpretation of their results to an entirely presynaptic mechanism, with effects primarily on bipolar cells.

[REDACTED]

[REDACTED]

(3) Our pharmacology results showed that TPMPA increased EPSC recorded in COMS-AC (Figure 3e-f), in alignment with the reviewer's suggestion that TPMPA acts on bipolar cells. However, the observed reduction in IPSCs (Figure 3g-h) suggests a direct influence of TPMPA on COMS-AC dendrites. If the effect of TPMPA were exclusively mediated by bipolar cells, one would anticipate an increase in IPSCs rather than a decrease. The rationale behind this expectation is that TPMPA would increase bipolar cell activity, consequently enhancing glutamate release from the bipolar cell to an AC that connects the bipolar cell to COMS-AC (bipolar cells→AC→COMS). Such an increased glutamate release would enhance the activity of this AC, resulting in increased GABA release from it and subsequently an elevated IPSC in COMS-AC."

(4) Unfortunately, specific antibodies targeting GABA_A are not commercially available, making it hard for us to demonstrate the expression of GABA_A in COMS-AC by immunohistochemistry.

[REDACTED]

4. The characterization of the HD2 cell as a non-object motion sensitive cell seems somewhat inaccurate. In the paper that characterized this cell type (Jacoby and Schwartz, J Neurosci 2017), this cell type responds better to differential motion than to global motion (though not as strongly to either than to a “center only” visual stimulation). The authors should clarify in the discussion how their stimuli and results relate to the original characterization of this cell type and consider changing their description of the cell type to be a more accurate description of this cell type’s preferences.

We appreciate the reviewer’s comment. In response, we have re-designated this cell as HD2p-RGCs (putative HD2-RGC). We also included additional sentences in the result section (page 8) and discussion section (page 13) to elaborate its relationship with HD2-RCC in Jacoby and Schwartz, J Neurosci 2017. It’s worth noting that the stimulation protocol used in our study has been widely used by many laboratories studying OMS, including studies by Zhang et al., PNAS 2012; Krishnaswamy et al., Nature 2015; Kim et al., eLife 2015; and Hsiang et al., eLife 2017. We do believe that establishing standardized stimulation protocols is important within the field.

Minor comments.

1. In the first paragraph of the introduction, the authors discuss the “current view” that retinal strata are specialized for encoding different stimulus properties. I was not aware that this was a generally held view. The authors should cite their source for this.

We have removed this sentence and replaced it with an acknowledgment of our limited knowledge regarding the functions of ACs. Thank you.

2. The authors do not mention the role of upstream bipolar cells in their introduction or discussion of mechanisms for object motion sensitivity, even though bipolar cells may play an outsized role in setting up OMS in the retina (see Gaynes et al, Nature Communications 2022).

We have included the role of bipolar cells in the introduction.

3. At the end of the first paragraph of the results, the authors mention several amacrine cell types from different classification systems that may correspond to the types identified with the intersectional strategy. Several acronyms are not defined in these sentences and the different classifications are not well-identified (one is EM and the other is transcriptomic). Displaying the EM data next to the cells identified using the intersectional strategy would be helpful to see in a figure.

We defined SBEM and single-cell RNA sequencing, and we added SBEM types to Figure 1f, 1h.

4. The authors should include the GCaMP isoform in all mentions of this in the manuscript – whether it is ‘6f’ or ‘6s’.

As suggested, we have incorporated all the GCaMP isoforms into the manuscript.

5. The figures are confusingly laid out, with panels for similar measurements in random spatial locations. For instance, in Figure 4, the same recordings and summary data are made in HD2 and W3 cells but the panels are not in a column as they could be to facilitate comparison. In addition, several of the electrophysiology traces did not print well and seem to require further down-sampling for display.

Thank you for your valuable suggestions. We have down sampled the electrophysiology traces in Figures 5h and 7j. We have also attempted to reformat figures as suggested. However, due to the large amount of data we would like to include and the constraints of space within each Figure, re-arranging those panels in columns turned out to be challenging for this manuscript. We will certainly keep it in mind for our future manuscripts.

Reviewer #2 (Remarks to the Author):

The mammalian retina, like other brain regions, comprises >100 types of neurons. Highly specialized connectivity within and between types plays an important role in diversifying visual signals before leaving the optic nerve. This article by Jo and colleagues utilizes a powerful intersectional genetic approach for exploring connectivity between retinal ganglion cells and retinal interneurons. They identify 2 interneurons for which very little is known: CK2-AC1 and CK2-AC2, and are able to label one or both with various reporters and ChR2. Receptive field properties were pursued with selective GCamp6

expression in CK2-AC1. Spots of light revealed a preference for light decrements (OFF), and drifting grating revealed a strong preference for local movement over global movement. Voltage clamp recordings showed that EPSCs were reduced by global motion whereas IPSCs were enhanced. TPMPA, a GABA_AR antagonist, eliminated this effect. The authors then compare activity of two co-stratifying RGCs in response to local and global motion: the UHD and HD2. The UHD has a strong preference for local motion whereas the HD2 responds similarly to both local and global motion. ChR2 expression in CK2-AC1 revealed direct glycinergic connections between cells, and PSAM expression and exogenous modulation revealed a stronger response to local motion. Furthermore, the authors use their intersectional strategy to identify and modulate a wide-field amacrine cell type that inhibits the HD2, via GABA_ARs, during global motion. This is an impressive set of results, both technically and scientifically, and they should appeal to both retinal specialists and neuroscientists in general.

I see only two deficits in the study: 1) lack of identification of the WF AC that inhibits CK2-AC1, and 2) lack of RF field mapping for decrement steps, which the cells strongly prefer. Amacrine cell connectivity is notoriously difficult to identify and study, so the progress made here is, in my opinion, sufficient for publication even in the absence of WF AC identification. Im not really sure why the cells were not probed with decrement steps (from a mean), thus the Off responses shown here are really rebound responses.....this should be clarified explicitly in the text.

We are grateful to the reviewer for this critical suggestion. In response, we have incorporated Figures 2f and 2g to show the receptive field mapping for decrement steps. The receptive field associated with 'OFF' stimuli closely resembles that of 'ON' stimuli, with the peak amplitude reduced by half. The difference could likely be attributed to the contrast used, as 'OFF' stimuli transition from gray to dark, representing half the contrast range compared to 'ON' stimuli, which shift from dark to light.

Minor

1) I find the CK2-AC1(COMS-AC) notation to be a burdensome and a little confusing. Why use two names here? I suggest defining something upfront, and simplifying throughout the ms.

We thank the reviewer for this suggestion. As suggested, we have adopted the singular name "COMS-AC" throughout the entire manuscript.

2) In a few places the authors say that the CK2-AC1 encodes OMS, but I would argue this is incorrect. The CK2-AC1 is object motion sensitive, and encodes information about local object motion.....it doesn't really encode OMS.

We would like to thank the reviewer for bringing to our attention to this inaccuracy in our description. We have made revisions, "COMS-AC is sensitive to object motion", in these instances.

3) p.10 Redundant sentence.....To examine whether TH2-AC makes synaptic connections with HD2-RGC, we expressed ChR2 in TH2-AC while labeling HD2-RGC with YFP in TH-Cre;Ai32;W3-YFP mice. To investigate the presence of synaptic connections between TH2-AC and HD2-RGC, we utilized optogenetics by expressing ChR2 in TH2-AC while labeling HD2-RGC with YFP in THCre; Ai32;W3-YFP mice (Fig. 7A).

Thank you. We have removed the second sentence.

4) p.10 '0.16mV/mm²'I believe you meant to use mW instead of mV, please double check

We have changed it to 0.16mW/mm². Thank you.

5) p.12 OMS gets defined again, but is now referring to the encoding of non-OMS...this is confusing. I would argue this should be described as Object Motion Insensitive.

Thank you once again. We have replaced all instances of 'non-OMS' with 'OMS-insensitive'.

6) I'm a little surprised that the HD2-RGC has such a strong OFF response to a spot of light when CK2-AC1 has such a strong preference for OFF stimuli too. This seems like a feature that might also have cancelled, but it doesn't. I suggest adding a couple sentences to the discussion regarding this point.

This is a great point! COMS-AC contributes to the center inhibition of HD2p-RGC, as demonstrated in Figure 7c, where the silencing of COMS-AC results in increased HD2p-RGC spiking during small spot stimulation. However, this inhibition is not sufficient to completely suppress HD2-RGC spiking. We speculate that a comparatively low glycine release from COMS-AC (the cell was named nGnG1 due to the relatively low expression of Glyt1 compared with other glycinergic ACs), combined with its transient response to spot stimulation (Fig. 2h), results in a relatively weak and momentary inhibition in HD2p-RGC. This inhibition may not be potent enough to counteract the strong excitation experienced by HD2p-RGC during spot stimulation. We have added discussion to Page 14.

7) I'm not sure why all panels in Figure 4D are labelled with HD2-RGC. Seems unnecessary, and the image in Figure 4Dii, for example, doesn't have any labelled HD2s.

We originally labeled the panels with HD2-RGC to differentiate them from W3(UHD)-RGC in Figure 4j. As suggested, we have removed the labeling from all panels.

Small note: This reviewer really appreciates the control experiment in Figure 5. Sometimes exogenous compounds have unintended effects, and this was a clever way to control for this issue.

Reviewer #3 (Remarks to the Author):

Summary

This goal of this paper is to understand how signals from functionally distinct interneurons are mixed on physically proximate synaptic targets to generate different responses. The authors use the retina as a model and take aim at a pair of ganglion cells whose dendrites overlap extensively with amacrine cells (VG3 and CAII) whose signals are vital for object motion selectivity. One ganglion cell, called W3 sums input from VG3 and CAII to become object motion selective –it responds when motion in its center leads or lags motion in its surround. The other cell, called HD2, is the focus of this manuscript –its dendrites are entangled with VG3, CAII, and W3, but it does not show object motion selectivity. Using a series of elegant genetic experiments and electrophysiological analyses, the authors learn that HD2 receives inhibition from a newly identified AC type, CK2-AC1, which silences this RGC's object motion responses.

Explaining RGC visual responses in terms of their amacrine and bipolar inputs is a major, and as yet,

incomplete goal. We know a great deal about direction selective retinal circuits, but know very little about any other retinal circuits. The results in this study reveal connectivity in a new circuit and illustrate a novel finding –that distinct amacrine-RGC connectivity patterns within a retinal neuropil layer create distinct visual responses. Specifically, the authors show a pair of circuits occupying the same retinal neuropil layer differ in their preference for local motion because one of these circuits (HD2) receives input from a novel amacrine type (CK2). There are also important tools devised here –the access to CK2-ACs is likely to be important to the retina field while the TITL-PSAM line is likely to be useful to neuroscientists in general. I thought the experiment using PSEM/PSAM to 'unmask' local motion responses on HD2 RGCs was particularly revealing.

I liked this study and the authors are to be commended on a tough but revealing series of experiments. However, there were many typographical errors, nomenclature errors, and missing pieces of data which should be addressed.

Major comments

1) W3-RGCs are concentrated in the ventral half of the mouse retina (bleckert et al., Zhang et al.), with a possible interpretation being that they were specialized to sense aerial predators. Given the finding that HD2-RGCs are non-OMS due to CK2-AC1, it would be interesting to know the distribution of these cells across the retina. It is probably too hard for HD2, but given the stains for CK2-ACs in Fig.1, I wonder if the authors could report the absolute density of these two types and their regional bias (ie: more in dorsal retina or ventral retina, etc). I imagine they have this data from the VGAT-Cre/Camk2a-tta/Ai93 cross with a ppp1r17 stain.

Thank you for your valuable suggestion. We have added Supplementary Figure 1a to show the distribution of the two AC types across the retina. However, we do want to acknowledge the limitation of the strategy, as in most cases, the intersectional strategy only allowed us to label subpopulations of each cell type. Consequently, the data we present may not provide a complete representation of the most accurate distribution patterns of each type.

2) The comparison between global and local motion is used to measure the contribution of certain kinds of input (ie: gabazine) or between RGCs (W3 vs HD2). The stimulus listed as 'local' would really be a differential motion stimulus according to the original definition of OMS (Olveczky et al., 2003). Local motion would be a stimulus presented to the receptive field center only. The follow-up paper describing HD2 (Jacoby et al. 2019) used a variant of this stimulus which moved a central texture in a different direction than the surround texture (fig9 of Jacoby et al.). Given that the authors believe that CK2 provides input to HD2, it would be nice to see its responses to this texture stimulus to register the results seen here with previous work.

To avoid any mislead, we have renamed this cell type as HD2p-RGCs (putative HD2-RGC) and included additional sentences in the result section (page 8) and discussion section (page 13) to elaborate its relationship with HD2-RCC described in Jacoby and Schwartz, J Neurosci 2017. Because the current manuscript primarily focuses on the function of COMS-AC, we would prefer to make changes involving cell type annotations rather than conducting the texture stimulus experiments. This preference is also influenced by the fact that our graduate student, Andrew Jo, completed his studies a couple of months ago and has since relocated to another city. It's also worth noting that, as we responded to reviewer #1 major point 4, our stimulation protocol has been widely used by many laboratories researching OMS,

including studies by Zhang et al., PNAS 2012; Krishnaswamy et al., Nature 2015; Kim et al., eLife 2015; and Hsiang et al., eLife 2017. And we think it would be more efficient for the field to employ standard protocols whenever possible to enable more direct comparisons between different studies.

3) Are the responses of the two dendritic strata of CK2-AC1 the same for local, global, and differential stimuli? I assume both data were grouped for subsequent experiments, but it would have been nice to see a bit more analysis of this dataset that justifies that recordings from different strata can be combined. If this was not done, then how was imaging depth standardized?

We appreciate the excellent suggestion from the reviewer. In response, we have analyzed the responses at both strata and have presented the data in Fig. 4b and 4c.

4) It is cool that CK2-AC1 blunts the local motion response on HD2, but as the authors mention in the discussion, the 'feature' that HD2 responds best to is unknown. As discussed, it seems as though the HD2-RGC does not respond well to any stimuli. I think this discussion paragraph can be improved.

We agree with our reviewer. Unfortunately, understanding the 'feature' that HD2 best responds to requires a separate project. Given our current limitations in knowledge, we find it challenging and risky to delve further into this discussion at this time. However, we do want to convey that exploring the function of HD2-RGC remains one of our future directions.

5) The data in Fig6G and J is interesting and show that inhibiting the CK2-AC1 causes HD2-RGCs to become sensitive to local motion. As written in the introduction, VG3-ACs give rise to the local (and object) motion response of W3 – does the presence of a local motion response in HD2 mean that it also receives input from VG3? From the diagram in Fig.8, it seems as though the authors do not believe this is the case. Why? A sentence or two on this point would help a reader synthesize these results with the previous work on object motion.

6) The authors make a strong case that CK2-AC1 inhibition to HD2 blunts their local motion response. But given that VG3 provides glycinergic inhibition to suppressed-by-contrast RGCs, could it also inhibit HD2-RGCs? A sentence or two about this in the discussion would help.

We thank the reviewer for posing the two excellent questions. In response, we silenced VGlut3-AC in VGlut3-Cre;W3-YFP mice with AAV2(YF4)-smCBA-DIO-hM4Di (Supplementary Fig.9a), subsequently examining the effects on EPSC and IPSC in HD2p-RGC. As shown in Supplementary Fig. 9b-m, C21 (DREADD agonist) does not induce any discernible effects on EPSC or IPSC, whether during local or global motion. Consequently, our findings strongly indicate that VGlut3-AC does not play an active role in the OMS circuitry within HD2p-RGC. It is worth noting that these findings align with a recent 3DEM investigation which demonstrates the preference of VGlut3-AC to establish synaptic connections with W3 (UHD, 5ti) cells while avoiding interactions with HD2 (5so) cells (<https://www.biorxiv.org/content/10.1101/2023.07.03.547571v1>).

7) Some quantification of the synaptic latency in the optogenetic studies would have been nice to see.

We have re-plotted Fig. 5c and Fig. 8b, and added Supplementary Fig. 4 to show the synaptic latency. Our observations indicate that COMS-AC started to depolarize at 5.3 ± 0.2 ms after the onset of the blue light, displaying a rise time of 1.8 ± 0.1 ms from 20% to 80% of the peak amplitudes. Furthermore, HD2p-RGC IPSC started to rise ~ 7 ms after the onset of Chr2 activation, with a synaptic latency of ~ 2 ms following the onset of COMS-AC depolarization. These details have been included on Page 8.

8) There are many typographical errors in the text and figures. These need to be corrected.

We apologize for the typographical errors, and we have made corrections.

9) The methods are missing details about how LMPI was computed, how DRPs were computed, and other forms of analyses. These should be added. There's also little information about how fast the gratings were moving for local and global stimuli. Some sentences should also be written about how the receptive field centers of the CK2-ACs were found.

As suggested by the reviewer, we have included detailed information regarding LMPI, DRPs, and the receptive field centers in the 'Methods' section.

10) All throughout, I think the authors have mistaken W3 for UHD when it should LED – Jacoby et al., show that UHD and HD2 are not object motion sensitive. Related to point 3 above, it is the differential motion response (together with size/speed tuning that distinguish these 4 cells).

W3 corresponds to UHD rather than LED, which has been confirmed by Greg Schwartz. W3 possesses a relatively smaller dendritic field, approximately 100 μm in size, with dendrites extending across a broader stratification range, including OFF strata. Conversely, LED exhibits a larger dendritic field measuring 131 μm and a more restricted stratification pattern. In terms of light response, W3 displays an equal ON-OFF response, whereas LED is predominantly OFF-dominant in photopic conditions. On the other hand, W3 closely resembles UHD in both morphology and light responses. Greg Schwartz lab has characterized W3-YFP mice, and he has verified that W3 corresponds to UHD and HD2p-RGC corresponds to HD2-RGC. In fact, there are very few, if any, LED cells labeled in the W3-YFP mouse line.

Minor comments.

1) "These two types occurred at a ratio ranging from 4:1 to 9:1." What is the definition for this ratio, I assume CK2:CK1 but I am not sure. Can you clarify.

We have provided clarification by appending '(COMS-AC:CK2-AC2)' after '4:1 to 9:1'.

2) reference is missing: "Additionally, CK2-AC1 showed a strong resemblance to nGnG1 and expressed PPP1R17 immunostaining (Fig. 1K), which is also expressed in nGnG1 and nGnG3." I suspect this should be PMID: 32457074.

As suggested, we have added the reference.

3) "...while MAC 22 corresponds to nGnG3." What is MAC? By its position in the sentence, I'd guess it is a typo that should read CK2-AC2?

We apologize for the confusion. MAC stands for 'Müller glia-coupled amacrine cell,' as discovered in Grimes et al., J Neuroscience 2021. We have now included the definition for 'MAC'.

4) Fig.1E. There are two "ii" panels.

We changed the second panel to iii. Thank you.

5) what does this mean: "we investigated how CK2-AC1 responds to visual stimuli in different layers and what are its receptive field properties at each level." I think you mean "we investigated how visual responses were organized across the laminar depth of CK2-AC1 dendrites.

We appreciate the reviewer's suggestion, and we have implemented the corresponding changes. Thank you.

6) "These results suggest that the majority of inhibition to CK2-AC1 originates from GABAergic wide-field ACs and is mediated by GABAC receptors." There are several kinds of medium field ACs, how do the authors know it is wide? Some clarification needed here.

We agree with the reviewer and have removed "wide-field".

7) How is LMPI calculated? I could not find a section of the methods with this information,

The information was provided Figure Legend 4c. We have now also included it in the 'Methods' section.

8) "TH2-AC co-stratifies with CK2-AC1 and demonstrates varying kinetics in response to global and local motions 17." Motions should be "motion".

It has been corrected.

9) there are several references missing from this sentence: "The retinal inner plexiform layer (IPL) is divided into 10 strata, where ~13 types of bipolar cells and 60 types of amacrine cells (ACs) form excitatory and inhibitory synapses with more than 40 types of retinal ganglion cells (RGCs), resulting in their unique response properties."

The references have been incorporated into the introduction as suggested.

10) "These results suggest that the majority of inhibition to CK2-AC1 originates from GABAergic wide-field ACs and is mediated by GABAC receptors." There are several kinds of medium field ACs, how do the authors know it is wide? Some clarification needed here.

Thank you. We have removed "wide-field".

11) Figure 4G – "Glocal" should read "Global" I think.

It has been corrected. Thank you for bringing it to our attention.

12) "These findings suggest that GABAergic wide-field ACs provide both presynaptic and postsynaptic GABAC-mediated inhibitions to CK2-AC1 (Fig. 2N)." Inhibition is pluralized here and should be singular.

It has been corrected, thank you.

13) "These results suggest that during global motion, CK2-AC1 receives strong presynaptic and postsynaptic inhibitions that together produce an overall hyperpolarization." Inhibition is pluralized here and should be singular.

It has been corrected, thank you.

REVIEWERS' COMMENTS

Reviewer #1 (Remarks to the Author):

The authors have gone above and beyond to address my comments, and I am satisfied by their responses. This study characterizes a new amacrine cell type using an exciting combination of strategies. Most interestingly, pharmacogenetics is used to show how silencing this newly-identified amacrine cell type alters the receptive field properties of a downstream ganglion cell type, changing the ganglion cell's overall functional properties. The strategies presented here set a standard for how to approach characterizing the 60+ types of amacrine cells in the retina – an important goal for understanding retinal function.

Reviewer #2 (Remarks to the Author):

My concerns have been addressed sufficiently.

Reviewer #3 (Remarks to the Author):

The authors have satisfied all my concerns.

We sincerely appreciate the reviewers' enthusiasm for the manuscript. Their thoughtful comments and constructive suggestions have been invaluable in improving the quality of the work.